# Biosynthesis and engineering of the nonribosomal peptides with a C-terminal putrescine

Hanna Chen [1,3,4], Lin Zhong [2,4], Haibo Zhou[1,4], Xianping Bai[1], Tao Sun [1], Xingyan Wang[1], Yiming Zhao [1], Xiaoqi Ji[1], Qiang Tu[2], Youming Zhang[1,2] & Xiaoying Bian [1] ✉

The broad bioactivities of nonribosomal peptides rely on increasing structural diversity. Genome mining of the *Burkholderiales* strain *Schlegelella brevitalea* DSM 7029 leads to the identification of a class of dodecapeptides, glidonins, that feature diverse N-terminal modifications and a uniform putrescine moiety at the C-terminus. The N-terminal diversity originates from the wide substrate selectivity of the initiation module. The C-terminal putrescine moiety is introduced by the unusual termination module 13, the condensation domain directly catalyzes the assembly of putrescine into the peptidyl backbone, and other domains are essential for stabilizing the protein structure. Swapping of this module to another two nonribosomal peptide synthetases leads to the addition of a putrescine to the C-terminus of related nonribosomal peptides, improving their hydrophilicity and bioactivity. This study elucidates the mechanism for putrescine addition and provides further insights to generate diverse and improved nonribosomal peptides by introducing a C-terminal putrescine.

Nonribosomal peptide synthetases (NRPSs) are large multidomain enzymes that operate in an assembly line fashion to produce a broad variety of nonribosomal peptides (NRPs) with diverse bioactivities and functions[1–3]. Each module of NRPS consists of an adenylation (A) domain, a thiolation (T) domain (also as a peptidyl carrier protein (PCP) domain), and a condensation (C) domain. The A domain recognizes and activates the target amino acids at the expense of ATP. The adenylated intermediate is transferred to the phosphopantetheinyl arm tethered to a T domain. The C domain catalyzes peptide bond formation between sequential amino acids. Last, the additional thioesterase (TE) domain releases the mature peptide product via hydrolysis or cyclization[3,4].

Nonribosomal lipopeptides have fatty acyl chains at the N-terminus, which are introduced by a starter condensation (Cs) domain[5–7] or a fatty acyl ligase[8,9] frequently located at the initiation module of NRPS, and the fatty acyl chains of lipopeptides show significant effects on their biological activities and toxicities[10,11]. Interestingly, nonacylated products were found in some lipopeptide NRPS gene clusters[12–15], mainly from the deacylation of lipopeptides mediated by acylases or peptidases[16]. The typical acylase PvdQ from pyoverdine biosynthesis belonging to the Ntn-hydrolase family was characterized, which was responsible for the cleavage of the myristate or myristoleate moiety from the N-terminal amino acid[12]. To date, the reported acylases PvdQ play vital roles in siderophore maturation and quorum sensing, as well as detoxification, by cleaving acyl chains in different bacterial taxa[12,14,15].

In addition to the well-known diverse acyl units at the N-terminus of nonribosomal lipopeptides, they also possess unusual moieties

[1]Helmholtz International Lab for Anti-infectives, Shandong University–Helmholtz Institute of Biotechnology, State Key Laboratory of Microbial Technology, Shandong University, 266237 Qingdao, Shandong, China. [2]CAS Key Laboratory of Quantitative Engineering Biology, Shenzhen Institute of Synthetic Biology, Shenzhen Institute of Advanced Technology, Chinese Academy of Sciences, 518055 Shenzhen, China. [3]Present address: School of Medicine, Linyi University, Shuangling Road, 276000 Linyi, China. [4]These authors contributed equally: Hanna Chen, Lin Zhong, Haibo Zhou. ✉e-mail: bianxiaoying@sdu.edu.cn

assembled at the C-terminus beyond the prediction of biosynthetic assembly lines. The additional C-terminal moieties include the amino acid residues that might be mediated by the TE domain, such as bacillothiazols[17] and a threonine-tagged lipopeptide family[18–22], and diverse terminal amines, such as diamine putrescine (Put), spermidine (Spe), agmatine (Agm), phenylethylamine (Pea), tryptamine (Tra), and tyramine (Tya)[14,23–31]. Putrescine is ubiquitously and abundantly distributed in living organisms[32] and is used as a precursor of other polyamines, surfactants, siderophores, and agrochemicals[33,34]. Putrescine is fully protonated at physiological pH and is usually detected in the unmodified form, although Put is also modified for purposes of metabolic regulation and catabolism, such as N-acetylation and N-glutamylation, or for incorporation into specialized metabolites, such as N-methylation and N-hydroxylation[35]. Furthermore, the direct incorporation of unmodified Put into the C-terminus of several natural products was prevalent in some NRPs, which was widespread in *Burkholderiales*[14,23–25]. However, the biosynthetic mechanism of the Put moiety participating in the C-terminus of these NRPs remains elusive and controversial. At present, two speculations have been proposed based on the features of their NRPS biosynthetic gene clusters (BGCs): one was that the C-terminal C domain might catalyze nascent peptide chain condensation with Put, such as malleobactins[24] and bicornutin A[28], and the other was that a VibH-like protein elsewhere in the chromosome involved in the condensation of putrescine, such as crochelin[14].

Genome sequencing reveals that the microbial genome is a massive and untapped trove of new metabolites[36,37]. Efficient strategies for the genome-driven discovery of novel natural products by awakening silent BGCs have been developed, such as functional promoter insertion in native producers and heterologous expression[38,39]. In our previous study, we successfully activated several silent/cryptic NRP BGCs in *Burkholderiales* strains by in-situ promoter insertion using the efficient recombineering system Redαβ7029 from *Schlegelella brevitalea* DSM 7029[19,40–42], a potential chassis for gram-negative bacterial natural products[39,43]. Herein, we identified a class of linear dodecapeptides, glidonins A-L (1-12), by triggering an NRPS BGC11 in the DSM 7029 strain[40]. The glidonins feature diverse modifications at the N-terminus and a putrescine moiety at the C-terminus. The biosynthetic pathway of glidonins was elucidated via in vivo genetic manipulation and in vitro biochemical assays. Moreover, the putrescine moiety could be added into potential peptides to generate novel and improved non-natural NRPs by bioengineering the termination module of NRPSs.

## Results

### Characterization of the glidonin biosynthetic gene cluster

Our previous study showed that the silent NRPS BGC11 from *S. brevitalea* DSM 7029 was activated successfully by in-situ constructive promoter $P_{Apra}$ insertion using Redαβ7029 recombineering[40] and produced a series of products, designated glidonins here, in the activated mutant *S. brevitalea* DSM 7029 $P_{Apra}$-BGC11, as shown by comparative metabolic profile analysis[40] (Supplementary Fig. 1). Bioinformatic analysis showed that the glidonin (*gdn*) BGC included two essential core NRPS genes (*gdnA* and *gdnB*) and several additional genes (Fig. 1a and Supplementary Table 1). To determine the additional genes involved in the biosynthesis of glidonins, we inactivated each gene through Redαβ7029 recombineering[40]. The LC–MS profiles showed that deletion of the gene *gdnC* (AKJ30060.1), which encodes an ABC transporter ATP-binding permease[44], abolished all products, indicating its critical role in the efficient transportation of glidonins, while the other mutants failed to impact the production of glidonin (Supplementary Fig. 2). Thus, the core genes of the *gdn* gene cluster are *gdnA*, *gdnB*, and *gdnC*, spanning approximately 44 kb contiguous DNA region.

The first NRPS GdnA contains an initiation Cs domain condensing a fatty acid chain to the peptidyl backbone to form a lipopeptide[5]. Its three modules (modules 1–3) are predicted to activate Phe, Asn, and Pro by A domain specificity analysis[45] (Supplementary Table 9). The NRPS GdnB includes nine canonical modules (modules 4–12) activating Val, Phe, Pro, X (defined as an unknown amino acid by AntiSMASH), Val, Ala, Ala, Ser, and Ala, respectively[45], as well as an incomplete termination module 13 containing a C domain, a partial A domain (A*), a T domain, and a noncanonical TE domain with two putative TE domains ($TE_1$ and $TE_2$) identified by the presence of two conserved active motifs (GXSXG)[46] according to bioinformatics analysis[47] (Supplementary Figs. 19 and 20 and Supplementary Table 9). The absence of the N-terminal subdomain ($A_{core}$) and the Stachelhaus codes in the A* domain leads to the A* domain retaining only a C-terminal subdomain ($A_{sub}$), which means it fails to function in amino acid activation (Supplementary Fig. 19b, c). The A* domain shows similarity (Coverage: 190/409; Identities: 41%-45%) to other A domains of GdnB, but it is not possible to predict the specificity of the A* domain due to the absence of the Stachelhaus codes (Supplementary Table 15 and Supplementary Fig. 19d). This kind of A* is rare in nature besides the non-functional A* domain of the kutzneride assembly line, which is critical for the stand-alone A domain KtzN-mediated transfer of Glu[48]. Further investigation is needed to determine the function of unusual termination module 13 of the *gdn* BGC, and structural elucidation of glidonins could provide valuable insights.

### Structural elucidation of glidonins

Eight target products were purified from the fermentation of *S. brevitalea* DSM 7029 $P_{Apra}$-BGC11. Glidonin A (1, Fig. 1c, Supplementary Note 1) was obtained as a colorless oil with the molecular formula $C_{65}H_{98}N_{16}O_{14}S$ (HR-ESI-MS, *m/z* 680.3699 [*M* + 2H]$^{2+}$, calcd 1359.7242). Interpretation of its $^1H$, $^{13}C$, and DEPT NMR spectra (Supplementary Table 2, in MeOD-$d_4$) revealed that 1 was a dodecapeptide containing the following amino acid moieties: Met, Asn, two Pro, two Val, Phe, Trp, three Ala, Ser, and one Put. The sequence of the amino acids and the location of Put were established by detailed interpretation of the HMBC correlations from amide protons and adjacent carbonyl groups (Supplementary Fig. 3, in DMSO-$d_6$). The NMR data of glidonin B (2) and glidonin D (4), whose molecular formulas were shown to be $C_{65}H_{98}N_{16}O_{15}S$ and $C_{66}H_{98}N_{16}O_{15}S$ by HR-ESI-MS, were also quite similar to the NMR data of 1, except for the obvious chemical shift variation of the methionine sulfoxide [Met(O)] residue in 2 and an additional aldehyde group ($\delta_H/\delta_C$ 8.15/164.0) in 4 (Supplementary Tables 3 and 4). The Met(O) in 2 is isomer and, in fact, was observed as a mixture of two epimeric compounds (major: 2a, minor: 2b). The molecular formulas of glidonins F-J (6-10) were found by HR-ESI-MS to be $C_{75}H_{116}N_{16}O_{15}S$, $C_{77}H_{120}N_{16}O_{15}S$, $C_{77}H_{120}N_{16}O_{16}S$, $C_{78}H_{122}N_{16}O_{15}$, and $C_{79}H_{124}N_{16}O_{15}S$. A comparison of the NMR data of 6, 7, and 10 (Supplementary Tables 4, 5, and 6) with 1 and analysis of their 2D NMR spectra (Supplementary Fig. 3) revealed that a decanoyl in 6, a lauroyl in 7, and a myristoyl in 10 are connected to the amino group of the Met moiety. Similarly, an additional lauroyl in 8 is connected to the amino group of Met(O) compared with 2, which was supported by 2D NMR correlations (Supplementary Fig. 3). Compound 8 also existed as a mixture of isomers, and the NMR data for the major rotamer of 8 are summarized in Supplementary Table 5. The 1D NMR data (Supplementary Table 6) of 9 are also quite similar to those of 7 except for the replacement of Met in 7 by Leu in 9. The structural compositions of the above eight compounds were confirmed by HR-ESI-MS/MS fragmentation (Supplementary Figs. 9, 10c, 10d, 11c, 11d, 12, and 13). Due to low yields, the structures of glidonin C (3), glidonin E (5), glidonin K (11), and glidonin L (12) were characterized by detailed comparative analysis of HR-ESI-MS/MS fragmentation (Supplementary Figs. 10a, 10b, 11a, 11b, and 14).

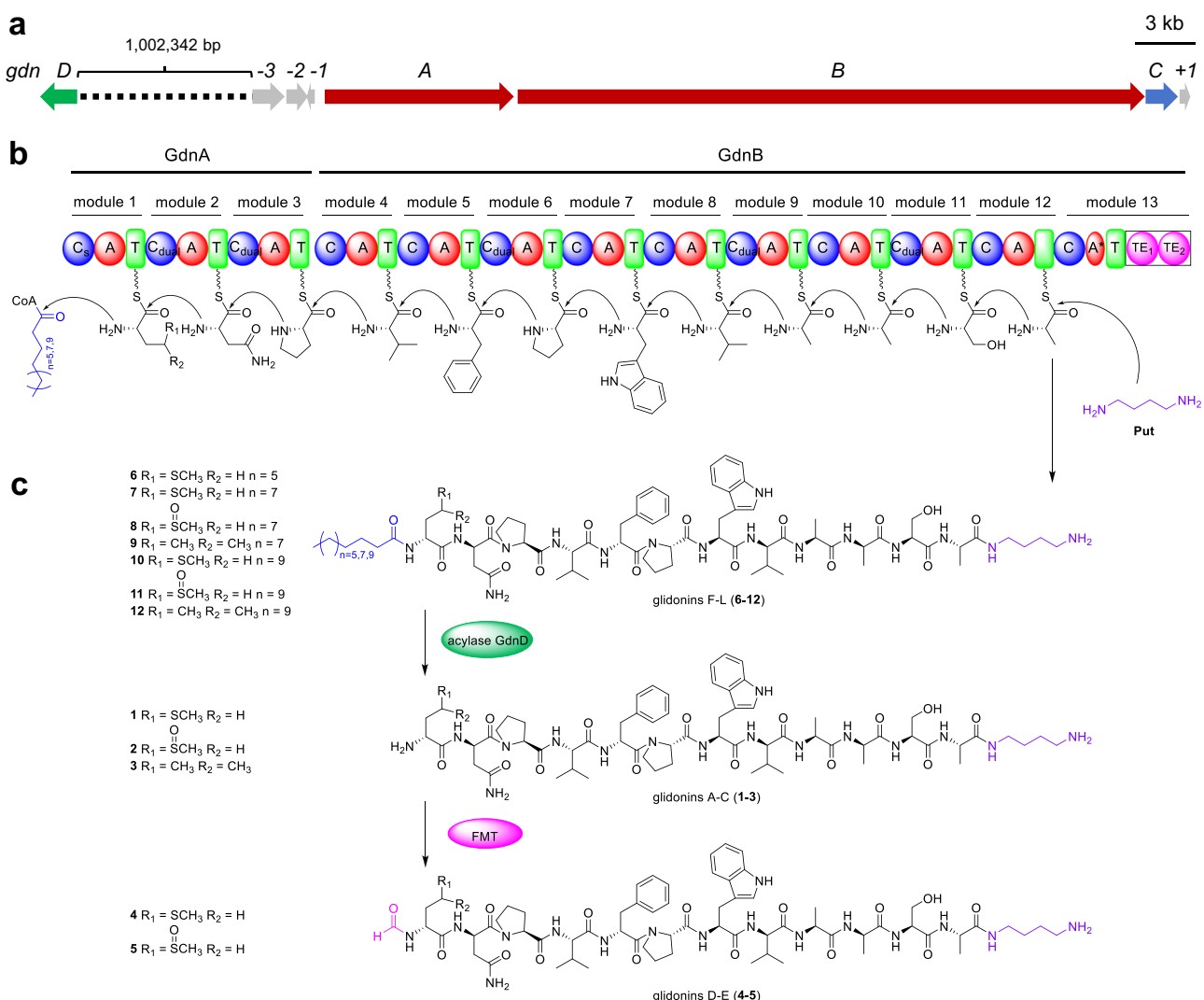

**Fig. 1 | The biosynthetic pathway of glidonins. a** Genetic organization of the *gdn* gene cluster. **b** Organization of modular enzymes involved in the biosynthesis of glidonin. **c** Structures and formation of glidonins A-L (**1–12**). FMT methionyl-tRNA formyltransferase, Cs starter condensation, C$_{dual}$ dual condensation (C/E), C condensation, A adenylation, T thiolation, TE thioesterase, A* partial adenylation.

The absolute configuration of the amino acid monomers in these compounds was determined by Marfey's analysis[49] combined with bioinformatic analysis of five dual C domains with epimerization and condensation activities[50] in the Gdn NRPS assembly line (Fig. 1b, Supplementary Fig. 23, and Supplementary Table 10). The results demonstrated that the amino acid constitutions of these compounds are D-Met[1](or D-Met(O)[1]/D-Leu[1])-D-Asn[2]-L-Pro[3]-D-Val[4]-D-Phe[5]-L-Pro[6]-L-Trp[7]-D-Val[8]-L-Ala[9]-D-Ala[10]-L-Ser[11]-L-Ala[12] (Fig. 1c).

### The formation of N-terminal diversity of glidonins

The final products of the Cs domain-embedded *gdn* gene cluster should be lipopeptides, but the nonacylated products (**1**, **2** and **3**) were identified here. To test whether **1-3** were derived from the incomplete function of the Cs domain, we inactivated the Cs domain in-situ by two patterns, which failed to produce any glidonins, indicating that the Cs domain was required for the biosynthesis of all glidonins (Fig. 2a). Therefore, we hypothesized that a deacylase is responsible for the deacylation of lipopeptide glidonins as a post-assembly modification. The genome of *S. brevitalea* DSM 7029 contains three PvdQ-like acylase enzymes (AKJ26698.1, AKJ30942.1, AKJ29580.1). The deletion of AKJ30942.1 (named GdnD), which showed 68% similarity to PvdQ and was far away from the *gdn* BGC (1,002,342 bp), abolished the

production of **1-5**, as shown by HPLC–MS analysis (Fig. 2b and Supplementary Fig. 5d). Subsequently, we used the cell lysate containing acylase GdnD (85 kDa) to test its deacylation function because we failed to obtain the soluble protein (Supplementary Fig. 5a). The nonacylated products **1** and **2** were detected in the reaction mixture containing GdnD incubated with **7** and **8** for 4 h, 16 h, and 27 h, respectively, but not in the boiled cell lysate (Fig. 2c). Therefore, GdnD is responsible for the removal of the fatty acid chain as a post-assembly modification in the biosynthesis of **1** and **2**. In addition, GdnD was also functional for other lipopeptides with mid- or long acyl chains and lipopeptides with the L-/D- configuration of the first amino acids and different lengths of peptide chains, exhibiting some substrate diversity (Fig. 2e, f and Supplementary Note 2). Notably, we detected formylated **1** (**4**), **13a** (**13b**) and **14a** (**14b**) in the reaction mixture of the GdnD assay (Fig. 2d, g and Supplementary Fig. 22), indicating that the formylation could be catalyzed by a formyltransferase from *Escherichia coli*, and **4** should be derived from tailoring formylation of **1**. The in vivo gene deletion and in vitro assays allowed us to determine that **1** was formylated by methionyl-tRNA formyltransferase to form **4** (Supplementary Note 3).

The glidonins contain different fatty acyls and the first amino acid residues, which is generally due to the substrate selectivity of the Cs

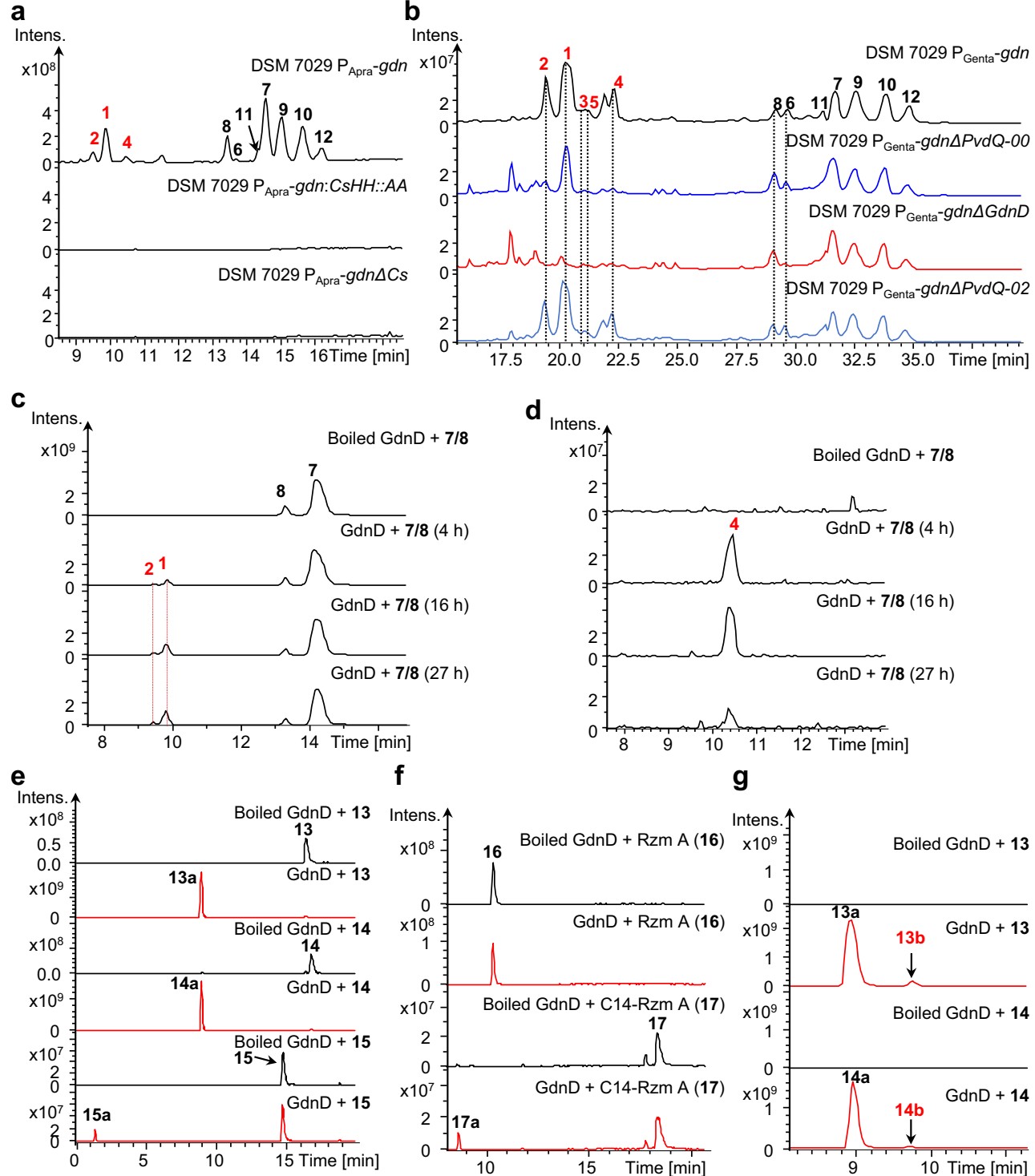

**Fig. 2 | Identification of Cs domain and acylase PvdQ enzymes. a** LC−MS analysis of crude extracts at EIC 650-800 of activated mutant DSM 7029 P(Apra)-*gdn* and two inactivated mutants of Cs domain, DSM 7029 P(Apra)-*gdn:CsHH::AA* and DSM 7029 P(Apra)-*gdnΔCs*. **b** HPLC−MS profile of crude extracts of activation mutant DSM 7029 P(Genta)-*gdn*, and three deletion mutants of acylases PvdQ. PvdQ-00: AKJ26698.1, GdnD: AKJ30942.1, PvdQ-02: AKJ29580.1. **c**−**g** Biochemical characterization of the acylase GdnD-catalyzed reactions by LC−MS analysis. **13**: lauroyl-D-Met-D-Asn-L-Pro-L-Val-D-Phe (EIC 789 [*M* + H]⁺), **13a**: D-Met-D-Asn-L-Pro-L-Val-D-Phe (EIC 607 [*M* + H]⁺), **14**: lauroyl-L-Met-D-Asn-L-Pro-L-Val-D-Phe (EIC 789 [*M* + H]⁺), **14a**: L-Met-D-Asn-L-Pro-L-Val-D-Phe (EIC 607 [*M* + H]⁺), **15**: lauroyl-D-Met-D-Asn (EIC 446 [*M* + H]⁺), **15a**: D-Met-D-Asn (EIC 264 [*M* + H]⁺), **16**: Rzm A (Rhizomide A) (EIC 732 [*M* + H]⁺), **17**: C14-Rzm A (C14-Rhizomide A) (EIC 900 [*M* + H]⁺), **17a**: deacylation of C14-Rzm A (EIC 690 [*M* + H]⁺), **4**: glidonin D (EIC 693.2 [*M* + 2H]²⁺), **13b**: formyl-D-Met-D-Asn-L-Pro-L-Val-D-Phe (EIC 633 [*M* + H]⁺), **14b**: formyl-L-Met-D-Asn-L-Pro-L-Val-D-Phe (EIC 633 [*M* + H]⁺).

domain and $A_1$ domain in initiation module 1. The formation of an unusual Met(O) of **2** might be a Met oxidized derivative during the isolation process from facile oxidation of the sulfide in Met of **1**. However, **1** and **2** were detected in the fermentation broth, and purified **1** was dissolved in MeOH for 3 days without changing to **2**, suggesting that the Met(O) residue was not generated by a tailoring reaction that took place after peptide formation. We assumed that the $A_1$ domain might select the nonproteinogenic amino acid Met(O). To test this hypothesis, we constructed and obtained the soluble protein CsA$_1$T$_1$ from *E. coli* BAP1[51] (Supplementary Fig. 7a). The in vitro adenylation activity toward the $A_1$ domain, as determined by an ATP/PPi exchange assay[52], demonstrated that the $A_1$ domain could indeed select L-Met(O), but the preferred substrate was L-Met (Supplementary Fig. 6a). The successful loading of substrates L-Met(O), L-Met, and L-Leu onto the $T_1$ domain was confirmed by using A$_1$T$_1$ and CsA$_1$T$_1$ proteins (Supplementary Fig. 24). We also tested the selectivity of Cs using lauroyl-CoA (C12-CoA) and myristoyl-CoA (C14-CoA) as donor substrates and L-Leu, L-Met, and L-Met(O) as acceptor substrates. Lauroyl products **18-20** and myristoyl products **21-23** were detected in vitro (Supplementary Fig. 6b, c), demonstrating that Cs could incorporate lauroyl and myristoyl moieties into the peptide assembly, consistent with the in vivo results. Therefore, the diverse substrates of initiation module 1 might be employed as an exchange element for the combinatorial biosynthesis of NRPS to generate structural diversity in pharmacologically relevant compounds.

## Assembly of the putrescine moiety

Genetic and biochemical analyses of the assembly of the Put moiety at the C-terminus of nonribosomal peptides remain elusive. Comparing BGCs encoding nonribosomal peptides with a C-terminal Put moiety, we found that the C-terminal multi-domains of the last module in the different NRPS assembly lines were different, such as C-T di-domains, C-A-T-TE domains, and C domain (Fig. 3a and Supplementary Fig. 8). The function of termination module 13 (C$_{13}$A*T$_{13}$TE) of glidonin was investigated here to elucidate the mechanism of Put incorporation into the peptide chain. We first performed site-directed mutagenesis to inactivate the conserved active sites of the C$_{13}$, T$_{13}$, and TE domains in vivo (Fig. 3b). The metabolic profiles showed that the inactivation of the C$_{13}$ domain or T$_{13}$ domain abolished glidonins, and the mutants $\Delta$TE$_2$(S10624A) and $\Delta$TE$_{1/2}$(S10391A/S10624A) reduced the yields of glidonins, but the mutant $\Delta$TE$_1$(S10391A) had no obvious influence on the yields of glidonins compared with that of the wild type (Fig. 3c, d). Therefore, the C$_{13}$ domain and T$_{13}$ domain on module 13 were essential for glidonin biosynthesis in vivo, and the TE domain seemingly was not involved in the release of final products from the NRPS assembly line due to the presence of the final products in the mutants of the TE domain.

To gain more insight into the assembly of the Put moiety in glidonin biosynthesis by biochemical assays, we cloned and expressed eight NRPS fragments of GdnB, A$_{12}$T$_{12}$, A$_{12}$T$_{12}$C$_{13}$, A$_{12}$T$_{12}$C$_{13}$A*, A$_{12}$T$_{12}$C$_{13}$A*T$_{13}$, A$_{12}$T$_{12}$C$_{13}$A*T$_{13}$TE, A$_{12}$T$_{12}$C$_{13}$A*T$_{13}$TE$_{Mut1}$(S10391A), A$_{12}$T$_{12}$C$_{13}$A*T$_{13}$TE$_{Mut2}$(S10624A), and A$_{12}$T$_{12}$C$_{13}$A*T$_{13}$TE$_{Mut1/2}$(S10391A/ S10624A) (Supplementary Fig. 7b, c). All soluble proteins were obtained except A$_{12}$T$_{12}$C$_{13}$ and A$_{12}$T$_{12}$C$_{13}$A*T$_{13}$TE$_{Mut1/2}$, hinting that A* and the two conserved Ser residues of TE may be involved in the correct folding of the protein A$_{12}$T$_{12}$C$_{13}$A*T$_{13}$TE. Moreover, the importance of the T$_{13}$ domain in maintaining protein structural stability was demonstrated, as the purified A$_{12}$T$_{12}$C$_{13}$A* protein was relatively unstable and prone to precipitation after thawing. An ATP/PPi exchange assay[52] was conducted to evaluate the activity of the six mutants. A$_{12}$T$_{12}$C$_{13}$A*T$_{13}$ and A$_{12}$T$_{12}$C$_{13}$A*T$_{13}$TE maintained the same bioactivity toward substrate L-Ala as the control protein A$_{12}$T$_{12}$, but the others showed decreased activity of A$_{12}$ (Fig. 4a), which might be due to the instability or incorrect folding of A$_{12}$T$_{12}$C$_{13}$A*, A$_{12}$T$_{12}$C$_{13}$A*T$_{13}$TE$_{Mut1}$(S10391A), and A$_{12}$T$_{12}$C$_{13}$A*T$_{13}$TE$_{Mut2}$(S10624A).

Based on the results of the ATP/PPi exchange assay, we divided these proteins into two groups: group 1 included A$_{12}$T$_{12}$C$_{13}$A*T$_{13}$, A$_{12}$T$_{12}$C$_{13}$A*T$_{13}$TE and the negative control A$_{12}$T$_{12}$, and group 2 included A$_{12}$T$_{12}$C$_{13}$A*T$_{13}$TE$_{Mut1}$(S10391A) and A$_{12}$T$_{12}$C$_{13}$A*T$_{13}$TE$_{Mut2}$(S10624A), in the subsequent condensation assay to remove the impact of L-Ala$_{12}$ loading on the yield comparison of T$_{12}$-linked Ala condensing with Put.

Afterward, we incubated substrates L-Ala and Put with the six proteins for the condensation assay (Fig. 4b). The target product **24a** was observed in all reactions, including the reaction of protein A$_{12}$T$_{12}$ with relatively low yield, which indicated that the free amino of Put could attack the carbonyl of T$_{12}$-linked Ala to form Ala-Put (**24**). In group 1, the highest yield of **24a** was obtained in the reaction containing protein A$_{12}$T$_{12}$C$_{13}$A*T$_{13}$. The yield in A$_{12}$T$_{12}$C$_{13}$A* was higher than that in A$_{12}$T$_{12}$, although the reduced activity of A$_{12}$ to L-Ala showed that the C$_{13}$A*T$_{13}$ domains and C$_{13}$A* domains both greatly promoted the intermolecular condensation of T$_{12}$-linked Ala with Put in vitro (Fig. 4b). Notably, the protein A$_{12}$T$_{12}$C$_{13}$A*T$_{13}$TE showed a greatly decreased yield of **24a** compared with A$_{12}$T$_{12}$, revealing that the TE domain preserved the hydrolysis function and resulted in a reduced yield of intermediate T$_{12}$-linked Ala in vitro. In group 2, despite reduced activity toward the substrate L-Ala compared with A$_{12}$T$_{12}$C$_{13}$A*T$_{13}$TE and A$_{12}$T$_{12}$ in group 1, as mentioned above, the mutants A$_{12}$T$_{12}$C$_{13}$A*T$_{13}$TE$_{Mut1}$(S10391A) and A$_{12}$T$_{12}$C$_{13}$A*T$_{13}$TE$_{Mut2}$(S10624A) exhibited higher yields of the target compound (**24a**) than A$_{12}$T$_{12}$C$_{13}$A*T$_{13}$TE and A$_{12}$T$_{12}$ in the condensation test. These results further support the notion that the TE domain has hydrolysis activity, and this function is abolished when the typical Ser active site is mutated. In addition, the yield of **24a** in the reaction containing A$_{12}$T$_{12}$C$_{13}$A*T$_{13}$TE$_{Mut2}$ was higher than that in the reaction containing A$_{12}$T$_{12}$C$_{13}$A*T$_{13}$TE$_{Mut1}$ under similar activity for substrate L-Ala (Fig. 4a, b), which implied that mutation of the Ser site of the TE$_1$ and TE$_2$ domains both led to reduce the hydrolysis function of TE in vitro, and the effect of the Ser site mutation was more pronounced in the TE$_2$ domain than in the TE$_1$ domain, which was consistent with the results observed in vivo (Fig. 3c, d). Based on the aforementioned results, we can infer that the C$_{13}$ domain is directly involved in the intermolecular condensation of T$_{12}$-linked Ala and free Put. Moreover, the A* and T$_{13}$ domains are essential for protein folding and stability in vitro.

Combining the in vivo and in vitro results, it can be concluded that the C$_{13}$ domain selects Put as an intermolecular nucleophile for elongated peptide chain release, resulting in diamine incorporation at the C-terminus, and the A*T$_{13}$ di-domain is essential for stabilizing the NRPS structure. As we presented, the TE domain (both for TE$_1$ and TE$_2$) retains its hydrolysis function in vitro (Fig. 4b), which was further supported by the TE deletion test in vivo (Supplementary Fig. 32). The final products can still be obtained for the deletion of TE$_2$, and the product without Put can be produced for P$_{13}$-TE (deletion of C$_{13}$A*T$_{13}$ and insertion of the P$_{13}$ promoter for TE). Despite having a hydrolysis function, the TE domain does not directly participate in the release of final products. Instead, it can impact the folding or stability of the NRPS, as shown both in vivo and in vitro (Figs. 3d and 4a). This is supported by the observation that deletion of the TE domain and TE$_2$ significantly decreased the yield of the final products compared with the wild type (Supplementary Fig. 32). Moreover, the TE domain may also assist in the efficient operation of the full-length NRPS pipeline in vivo by hydrolyzing any stagnant by-products resulting from the malfunction of the Put release step, as we previously discussed regarding glidomide[41]. This was also supported by the observation of an intermediate (**9a**) without the Put moiety found in the crude extract of both wild-type and mutant strain with a constitutive promoter P$_{13}$[53] inserted in front of the TE domain (Supplementary Fig. 32).

The C$_{13}$ domain could condense free putrescine with T$_{12}$-linked Ala, suggesting that the acceptor binding pocket of the C$_{13}$ domain might be appropriate for Put. The docking between Put and the pocket of the C$_{13}$ domain protein predicted by AlphaFold2[54] showed that three

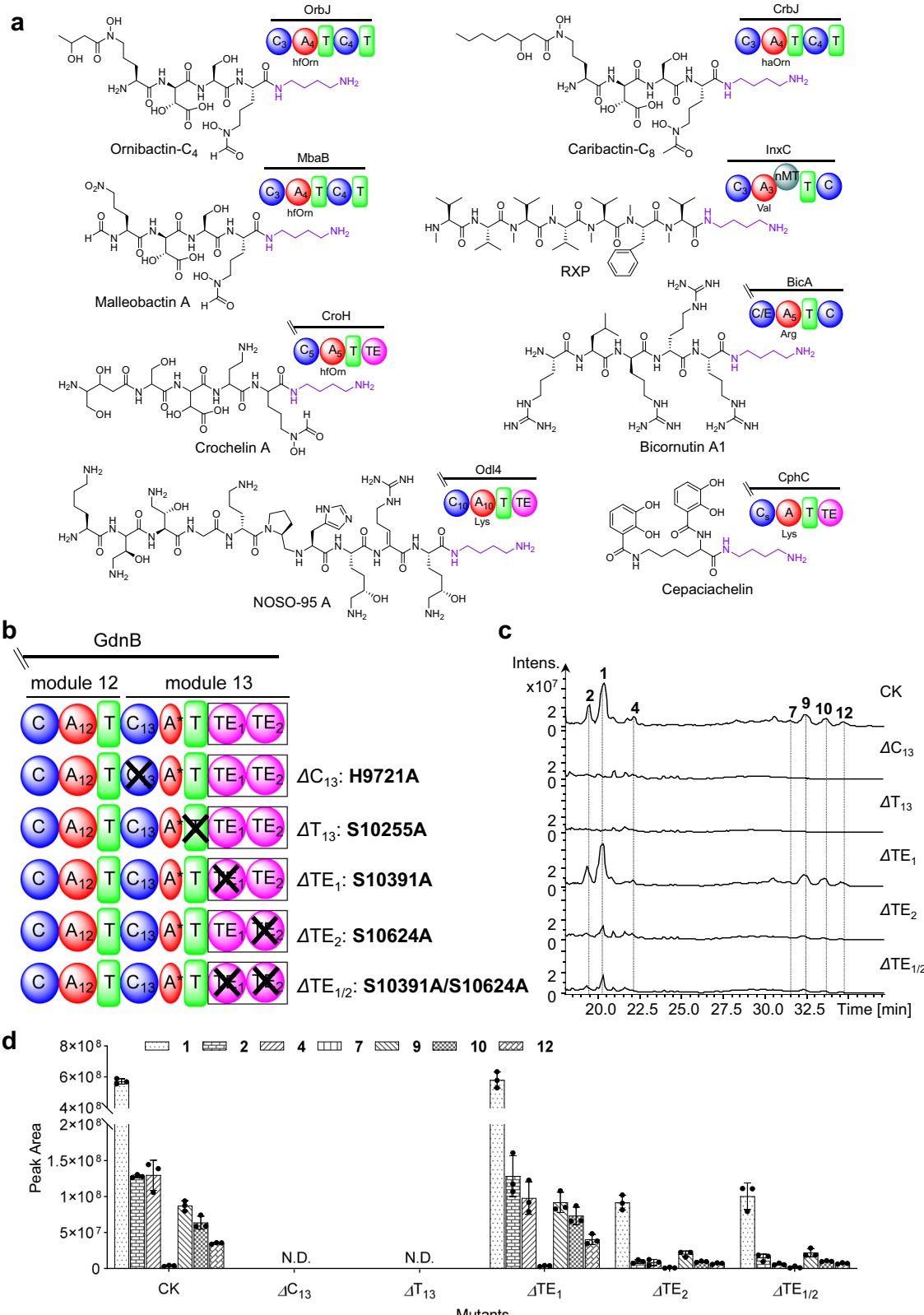

**Fig. 3 | Identification of the domains in charge of Put introduction in vivo. a** The nonribosomal peptides with a C-terminal Put and their termination modules of NRPSs. hfOrn: N⁵-formyl-N⁵-hydroxyornithine, haOrn: N⁵-acetyl-N⁵-hydroxyornithine, RXP: Rhabdopeptide/xenortide-like peptide. **b** Schematic diagram of the different mutations of module 13. **c** HPLC–MS chromatogram obtained from the crude extract of the culture of the different mutants. **1** ($m/z$ 1359.7 $[M + H]^+$), **2**

($m/z$ 1375.7 $[M + H]^+$), **4** ($m/z$ 1385.7 $[M + H]^+$), **7** ($m/z$ 771.4 $[M + 2H]^{2+}$), **9** ($m/z$ 762.4 $[M + 2H]^{2+}$), **10** ($m/z$ 785.4 $[M + 2H]^{2+}$), **12** ($m/z$ 776.4 $[M + 2H]^{2+}$). **d** Yield comparison of the compounds from the culture of the different mutants and wild type. CK indicates wild type, N.D. indicates no products. Data are presented as mean values ± SD, $n = 3$ biologically independent samples. Source data are provided as a Source Data file.

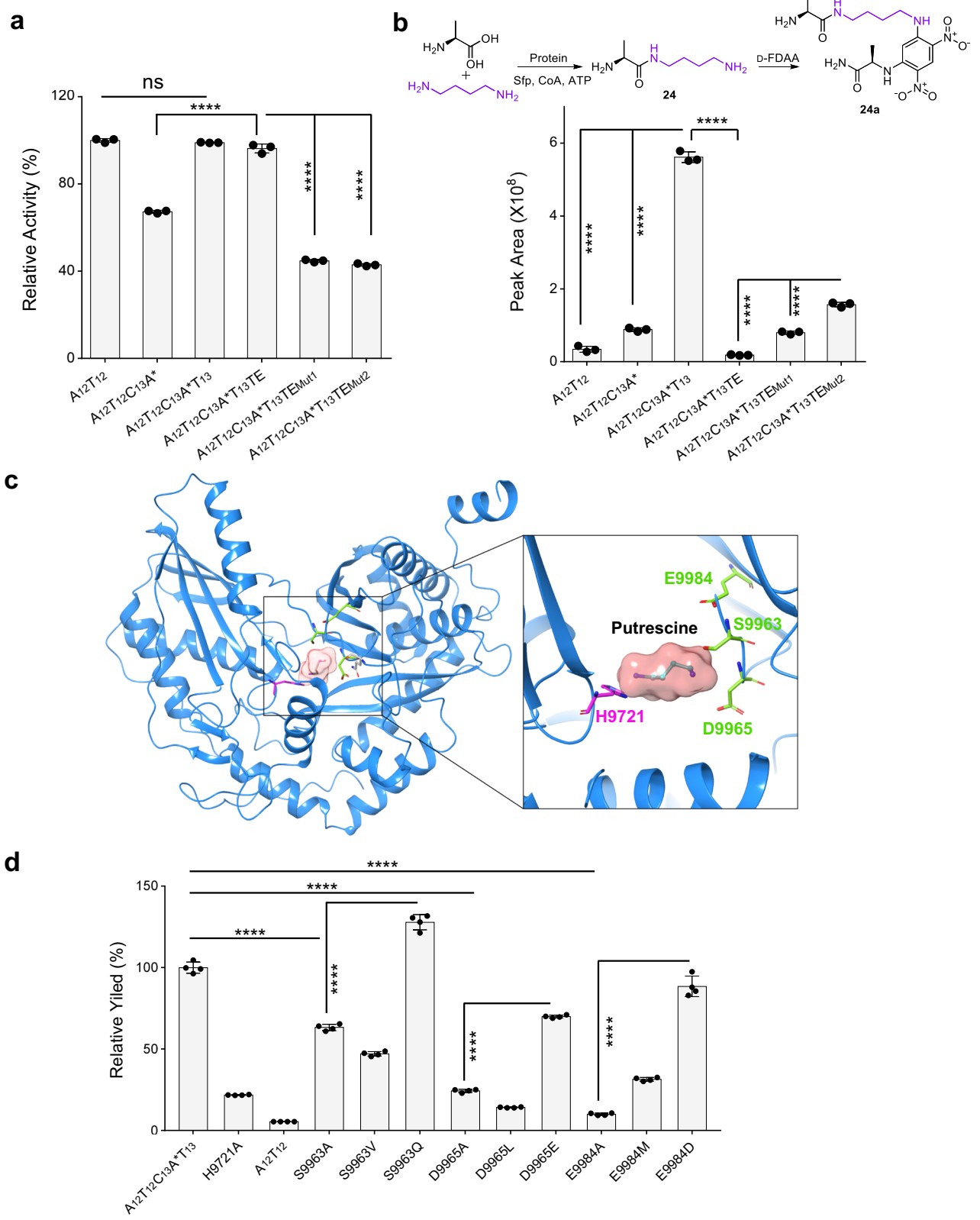

residues (S9963, D9965, E9984) of the $C_{13}$ domain, which were not conserved in other C domains, might be relevant to the substrate Put (Fig. 4c and Supplementary Fig. 17). Then, we constructed nine mutants of the $A_{12}T_{12}C_{13}A^*T_{13}$ protein to investigate the impact of these residues on the function (Supplementary Fig. 7d, e, f). The in vitro assays showed that the yield of **24a** from three mutants (S9963A, D9965A and E9984A) was greatly decreased, almost to that of mutant

$A_{12}T_{12}C_{13}A^*T_{13}$ (H9721A) and $A_{12}T_{12}$, at the premise of their similar activity toward the substrate L-Ala (Fig. 4d and Supplementary Fig. 16a, c). Moreover, the three mutants altered the substrate specificity of the $C_{13}$ domain for the substrates putrescine and 1,5-diaminopentane, resulting in a reduced catalytic activity toward putrescine and increased catalytic activity toward 1,5-diaminopentane (Supplementary Fig. 16b), indicating that these three crucial residues are located

**Fig. 4 | Identification of the Put assembly by in vitro biochemical experiments.** **a** The relative activity of $A_{12}$ domain from different proteins for the substrate L-Ala. The activity of $A_{12}T_{12}C_{13}A^*T_{13}$ for activation of L-Ala was quantified as a reference (100%). Two mutants are $A_{12}T_{12}C_{13}A^*T_{13}TE_{Mut1}(S10391A)$ and $A_{12}T_{12}C_{13}A^*T_{13}TE_{Mut2}(S10624A)$. TE contains $TE_1$ and $TE_2$. Data are presented as mean values ± SD, $n = 3$ biologically independent samples. **b** In vitro bioactivity assays of L-Ala condensing with Put mediated by the different proteins through comparing the yield of **24a** (**24** derivatized with D-FDAA). Data are presented as mean values ± SD, $n = 3$ biologically independent

samples. **c** The docking of $C_{13}$ domain and substrate Put, and zoom-in view of the conserved active site residue H9721 and key site residues S9963, D9965, E9984. **d** Yield comparison of the target product **24a** obtained from $A_{12}T_{12}C_{13}A^*T_{13}$, its nine mutants as well as negative control $A_{12}T_{12}$ and $A_{12}T_{12}C_{13}A^*T_{13}(H9721A)$. The yield of **24a** generated from the wild type $A_{12}T_{12}C_{13}A^*T_{13}$ was quantified as a reference (100%). Data are presented as mean values ± SD, $n = 4$ biologically independent samples. All $P$ values were determined by two-tailed unpaired $t$ test. **** $p < 0.0001$. Source data and the exact $p$ values are provided as a Source Data file.

within the binding pocket. The recovery of catalytic activity in the mutants with the original amino acid properties (S9963A vs. S9963Q, D9965A vs. D9965E and E9984A vs. E9984D) suggested the importance of the properties of these three residues (Fig. 4d). This indicated that these three residues might be key sites for substrate binding, especially E9984, which is located on the β-sheet of the $C_{13}$ domain. The impact of the length of the amino acid side chain on the product yield (S9963A vs. S9963V, D9965A vs. D9965L and E9984A vs. E9984M) suggested that it also created steric hindrance in the substrate binding pocket (Fig. 4d). In summary, the three residues were essential for interacting with substrate Put, and our mutation results indicated that engineering the substrate specificity of this releasing module is a viable strategy.

### Introduction of putrescine into NRPs by multi-domain swapping

Putrescine is also a precursor of some specialized compounds, such as industrial polyamide nylon 4,6[33] and siderophore alcaligin[55]. To clarify the function of the Put moiety of glidonins, we synthesized lipopeptide **10a** without the Put group. Compound **10** was easily soluble in water, methanol and acetonitrile, while compound **10a** was soluble only in dimethyl sulfoxide (DMSO), which indicated that the Put moiety could enhance the solubility of this lipopeptide. Moreover, the antitumor activities of **10a** against the tumor cell lines A549 and HepG2 were lower than those of **10**, demonstrating that the Put group might improve the bioactivities of lipopeptides (Supplementary Table 11). These results indicated that introducing the Put moiety into lipopeptides might have the potential to improve the chemical properties and bioactivities of lipopeptides.

To confirm this speculation, we tried to introduce the Put moiety into the assembly backbone of other lipopeptides by engineering the C-terminal module of NRPs. Two distinct types of lipopeptides, cyclic C8-rhizomide A[56] (C8-Rzm A, **28**) and linear holrhizin A[42,57] (**30**), from *Mycetohabitans rhizoxinica* HKI 454, were used as the target products. To obtain the desired products, the different fusion regions using interdomain linkers from *gdn* BGC to *rzmA*-R148A (*rzmA*\*) and *holA* BGCs were assessed. Considering that *rzmA*\* encodes the cyclic lipopeptide, the strategy of substitution was used for engineering *rzmA*\* (Fig. 5a, b), and we detected the expected mass of product **29** ($m/z$ 904.5882 $[M + H]^+$) in both mutants. Additionally, the yield of **29** in $C_{13}A^*T_{13}TE$ substitution of RzmA\*-M2 was higher than that in recombinant RzmA\*-M1 (Fig. 5c). We then purified and elucidated target derivative **29**, whose molecular formula was $C_{45}H_{78}N_9O_{10}$, by HR-ESI-MS (Supplementary Fig. 15a). NMR data and analysis of its 2D NMR spectra confirmed that **29** indeed included a Put moiety connected to the carboxyl group of Val (Supplementary Table 7).

For the *holA* BGC, we first explored four different fusion regions to replace C-terminal module 6 of HolA (Fig. 5d, e). Metabolite profiles showed that the expected product **31** ($m/z$ 774.5122 $[M + H]^+$) was detected in different recombinants, and the optimal yield of **31** was from $T_{12}C_{13}A^*T_{13}TE$ substitution of HolA-M1, which was better than the yield from $C_{13}A^*T_{13}TE$ substitution of HolA-M2 (Fig. 5f). Moreover, the yield of **31** from the $T_{12}C_{13}A^*$ substitution of HolA-M3 was higher than that from the $C_{13}A^*$ substitution of HolA-M4 (Fig. 5f), which indicated that the preservation of the $T_{12}$ domain in multi-domain substitution contributed to the increased production of derivative **31**.

Subsequently, the structure of derivative **31** was elucidated by HR-ESI-MS and NMR, which showed a Put moiety connected to the carboxyl group of Ala (Supplementary Table 8 and Supplementary Fig. 15b). To verify the effect of a Put moiety on the bioactivities of these compounds, two Put-lacking lipopeptides **29a** and **31a** were synthesized. Bioactivity assays showed that **29** and **31** indeed exhibited higher anti-inflammatory and antitumor activities than **29a** and **31a** (Supplementary Table 12 and Supplementary Fig. 29).

To further explore the minimal exchange units in charge of Put moiety assembly, six different exchange units were used to engineer the *holA* BGC (Fig. 5e). We found that four engineered strains produced another derivative (**32**) of holrhizin A, but HolA-M5 with $C_{13}$ domain substitution and HolA-M10 with $A^*T_{13}TE$ domain substitution did not (Fig. 5g), demonstrating that the minimal domain $C_{13}A^*$ was required for Put assembly in vivo, which was consistent with the aforementioned in vitro biochemical assays indicating that the $C_{13}$ domain was in charge of Put incorporation and that A\* might be very important for the structure of NRPS. The substitution sites of *rzmA*\* and *holA*, along with their corresponding amino acid sequences and protein structure characteristics, are summarized in Supplementary Fig. 25. During the substitution domain process, we also detected some short intermediates (**28a, 30a, 30b**) that were hydrolyzed in advance from the assembly line in different recombinants, as shown in Supplementary Fig. 30. In summary, the efficient substitution domains for the Put moiety engineered into NRPs by comprehensive analysis are the $C_{13}A^*T_{13}TE$ multi-domains, defined as the "Put unit", which has application prospects for improving the activity and enhancing the water solubility of target compounds in the bioengineering field.

### Introduction of other diamines into NRPs

We subsequently investigated the substrate tolerance of $C_{13}$ using other potential diamines and polyamides, including 1,7-diaminoheptane, 1,6-diaminohexane, 1,5-diaminopentane, 1,2-diaminocyclohexane, spermine, and spermidine, in addition to the native substrate Put in vitro. Only four corresponding compounds (**24a, 25a, 26a** and **27a**) were observed in the reaction mixtures of Put, 1,5-diaminopentane, 1,6-diaminohexane, and 1,7-diaminoheptane by LC–MS analysis. Moreover, the yield of **25a** was higher than that of **26a** and **27a** but lower than that of **24a** (Supplementary Fig. 31a), revealing that the available substrates of the $C_{13}$ domain were linear diamines, analogs of Put, and the longer lengths of diamines had a negative effect on the final product yield.

In addition, to extend the application of the "Put unit" in bioengineering, three usable diamines (1,7-diaminoheptane, 1,6-diaminohexane, and 1,5-diaminopentane) were added into the culture of the engineered strain *E. coli* GB05-MtaA:RzmA\*-M2. We found three corresponding desired derivatives **33** ($m/z$ 918.6032 $[M + H]^+$, calc. 918.6023), **34** ($m/z$ 932.6167 $[M + H]^+$, calc. 932.6179), and **35** ($m/z$ 473.8216 $[M + 2H]^{2+}$, calc. 946.6336) in the crude extracts in addition to the original compound **29**, as confirmed by HR-ESI-MS/MS fragmentation (Supplementary Fig. 31b, c, d). Bioengineering the "Put unit" can serve as a decent approach to enhance the diversity of NRPs with potential biological activities by feeding various diamines, and it can complement the Put binding pocket mutation method mentioned above.

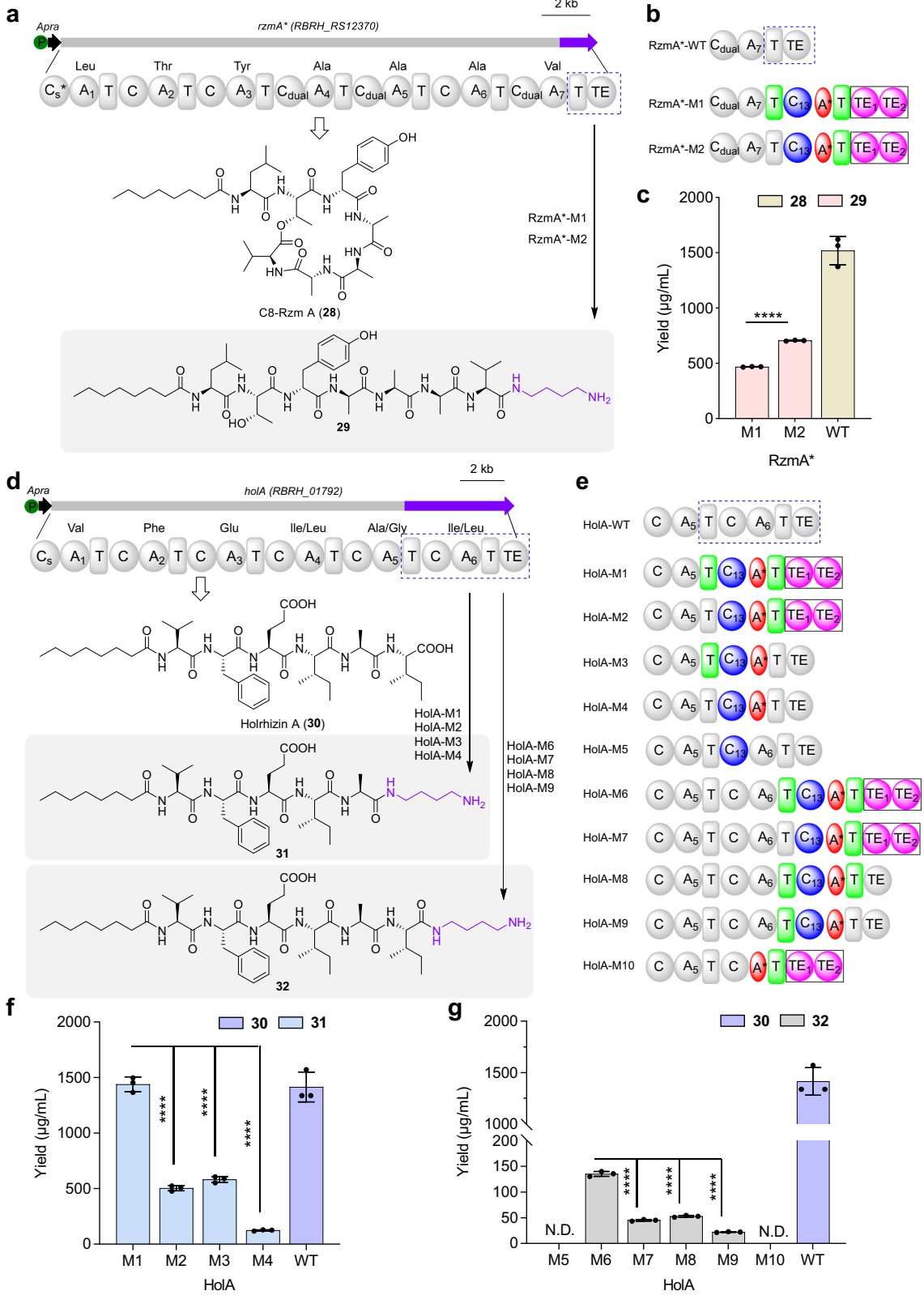

## The bioactivity and function of glidonins

The bioactivity of glidonins was screened by several assays (Supplementary Note 4). The cytotoxicity assay showed that lauroyl products **7** and **9** and myristoyl product **10** exhibited not only broad-spectrum cytotoxicity but also strong activities against several tumor lines (Supplementary Table 11). However, nonacylated compounds **2** and **3**

showed no activities, and **1** had moderate activities against the Kasumi and A549 cell lines.

To clarify the function of the deacylation step in glidonin biosynthesis for the host *S. brevitalea* DSM 7029, we compared the growth profiles and biofilm formation between the wild type (WT) and two mutants, DSM 7029 P$_{\text{Genta}}$-*gdn* and DSM 7029 P$_{\text{Genta}}$-*gdnΔgdnD*. The

**Fig. 5 | Introduction of putrescine moiety on the two NRPs by multi-domain swapping. a** The flow chart of engineering *rzmA** BGC and corresponding products. **b** A diagram of the different multi-domain substitutions of the C-terminus of RzmA-R148A (RzmA*). **c** Yield comparison of C8-Rzm A (**28**) and its derivative **29** between wild type (WT) and mutants. The absolute yield of related compounds was determined using the standard curve of **28** and **29**, respectively. Data are presented as mean values ±SD, *n* = 3 biologically independent samples. **d** The flow chart of engineering *holA* BGC and corresponding products. **e** A diagram of the different multi-domain substitutions of the C-terminus of holrhizin A. **f, g** Yield comparison of holrhizin A (**30**) as well as their derivatives **31** and **32** between WT and mutants. The absolute yield of related compounds was determined using the standard curve of **30a** and **31**, respectively. N.D. indicates no products. Data are presented as mean values ± SD, *n* = 3 biologically independent samples. All *P* values were determined by two-tailed unpaired *t* test. ****$p < 0.0001$. Source data and the exact *p* values are provided as a Source Data file.

growth profiles of the two mutants exhibited unusual growth curves because the mutants produced unknown pigments (black line and red line, respectively) (Fig. 6a and Supplementary Fig. 18a, b). Once pigment absorption was deducted (black and red dotted lines, respectively), the WT showed better cell growth than the two mutants, while the growth of the activation mutant DSM 7029 $P_{Genta}$-*gdn* was superior to that of the Δ*gdnD* mutant (Fig. 6a), which indicated that glidonins could inhibit the growth of DSM 7029 and that acylated glidonins might play a decisive role in this process. To directly evaluate the effect of acylated and nonacylated glidonins on wild-type DSM 7029, nonacylated compounds **1** and **2** and acylated compounds **7** and **10** were incubated with DSM 7029. The results revealed that the acylated compounds could obviously inhibit the growth of DSM 7029, but the nonacylated compounds had no effect (Fig. 6c). These results demonstrated that the deacylation of *gdn* BGC could be a detoxification mechanism for the native host DSM 7029.

The biofilm formation assays showed that DSM 7029 $P_{Genta}$-*gdn* and Δ*gdnD* increased biofilm formation compared to the WT, and the biofilm-forming capability of the activated mutant DSM 7029 $P_{Genta}$-*gdn* was higher than that of the Δ*gdnD* mutant (Fig. 6b). The colonies of the two mutants appeared smaller than those of the WT in the swarming motility assay (Supplementary Fig. 18d), which was consistent with the biofilm assays because swarming motility is often oppositely regulated and antagonistic to biofilm formation[58,59]. In addition, the colony morphology of the Δ*gdnD* mutant was dry and wrinkled compared to that of the WT and the activated mutant, and the cell morphology of the two mutants was sticky relative to that of the wild-type strain, as observed by scanning electron microscopy (SEM), due to the enhanced biofilm formation of the two mutants (Fig. 6e and Supplementary Fig. 18c). Additionally, acylated compounds **7** and **10** indeed increased biofilm formation of DSM 7029 wild type, but nonacylated compounds **1** and **2** had less effect on biofilm formation (Fig. 6d). Thus, acylated glidonins could improve biofilm formation by inhibiting the swarming motility of DSM 7029, leading to alterations in the colony morphology of DSM 7029. Based on the above results, acylated glidonins could affect the growth state of DSM 7029, including growth speed, biofilm formation, and pigment secretions, but nonacylated products could not.

## Discussion

The biosynthesis of Put, as the precursor for some natural products, has been intensively studied in various bacteria. However, the current issue of the assembly of free Put into natural products remained unclear. Only two similar studies about siderophores vibriobactins[60] and serratiochelins[61], whose free-standing amide synthases, VibH and SchH, incorporate free norspermidine/1,3-diaminopropane into the NRPS assembly line, were reported. In this work, we verified that the C-terminal $C_{13}A^*T_{13}$ domains embedded into the NRPS assembly line were essential for condensing Put with thioesterified intermediate products through in vivo and in vitro experiments. Therein, $A^*T_{13}$ domains may play a vital role in maintaining the correct folding of NRPS protein, and the $C_{13}$ domain is more similar to the individual amide synthases in condensation function. The second histidine in the "HHxxDG" motif also functions similarly to the typical C domain by playing a substrate positioning role (Fig. 4d, H9721A mutant vs. $A_{12}T_{12}$

mutant)[56]. In addition, the other representative NRPS assemble lines (Fig. 3a) possessing a C domain fixed into the C-terminus of termination module allowed us to deduce that the function of these C domains might be similar to that of amide synthases for Put introduction.

The addition of polyamines into natural products could enhance bioactivity and solubility, such as the anticancer F14512 linking a spermine chain to the epipodophyllotoxin core, showing priority efficiency in the pre- and clinical trials[62,63]. Here we also demonstrated the remarkable feature of the Put moiety to improve the solubilities and bioactivities of glidonins, and further supported by bioengineering the lipopeptides rhizomide and holrhizin to generate new-to-nature lipopeptides with increased activities. This finding provides further insights to biologists and chemists on the modification of potential lead compounds. The optimized "Put unit" could accomplish putrescine assembly, as well as the assembly of other diamines, which provides a decent strategy for expanding the diversity of lipopeptides by bioengineering in terms of combinatorial biosynthesis. In addition, the $A_1$ domain of *gdn* BGC showed good selectivity for the unusual nonproteinogenic amino acid Met(O) in vivo and in vitro, in addition to the proteinogenic amino acids Met and Leu, and can thus guide the introduction of unusual amino acids into NRPs through swapping of A domain or module to obtain diverse nonribosomal peptides.

The family of linear lipopeptides glidonins with diverse fatty acid chains (C10-C14) incorporates three different first amino acids. Interestingly, a high yield of nonacylated product from lipopeptide glidonins was found in the crude extract, which was processed postmodification by the characterized acylase GdnD. The acylase GdnD is clearly encoded outside the *gdn* BGC and deacylates a wide variety of lipopeptides. It is possible that the deacylation step might be a global detoxification or self-resistance mechanism for the native strain DSM 7029, similar to the role of the acylase Tem25 located inside the telomycin BGC[15]. Currently, the mechanisms of acylases in siderophore maturation and quorum sensing have been elucidated[12,14]. However, the natural function of several acylases, such as GdnD and Tem25, has yet to be resolved. Lipopeptides consist of a lipid moiety linked to a linear or cyclic oligopeptide, resulting in amphiphilic properties that confer enormous functional versatility. Their natural functions, some of which may be unique to the biology of the producing microorganisms, are involved in diverse ecological roles as biosurfactants and mediators of competitive advantage in interactions with coexisting organisms, as well as in biofilm formation[64,65]. The lipopeptides, acylated glidonins, not only increased the biofilm formation of DSM 7029 to improve its competitive advantage but also had negative effects on its growth. To balance this contradiction, DSM 7029 chooses to silence the *gdn* BGC under laboratory conditions and evolve the acylase GdnD. In addition, GdnD preferentially deacylates lipopeptides with long fatty chain lengths, which also have better bioactivities. This finding expands the knowledge of the self-protection for strains to adapt to the natural competitive environment.

In conclusion, the complex biosynthetic pathway of glidonin was elucidated, which starts with Cs domain to introduce fatty acids to the first amino acid, followed by condensation of the C-terminal Put by the $C_{13}$ domain of module 13, forming the final lipopeptide. The elucidation of the putrescine addition mechanism and the "Put unit" swapping strategy provided further insights into the engineering of diverse NRPs

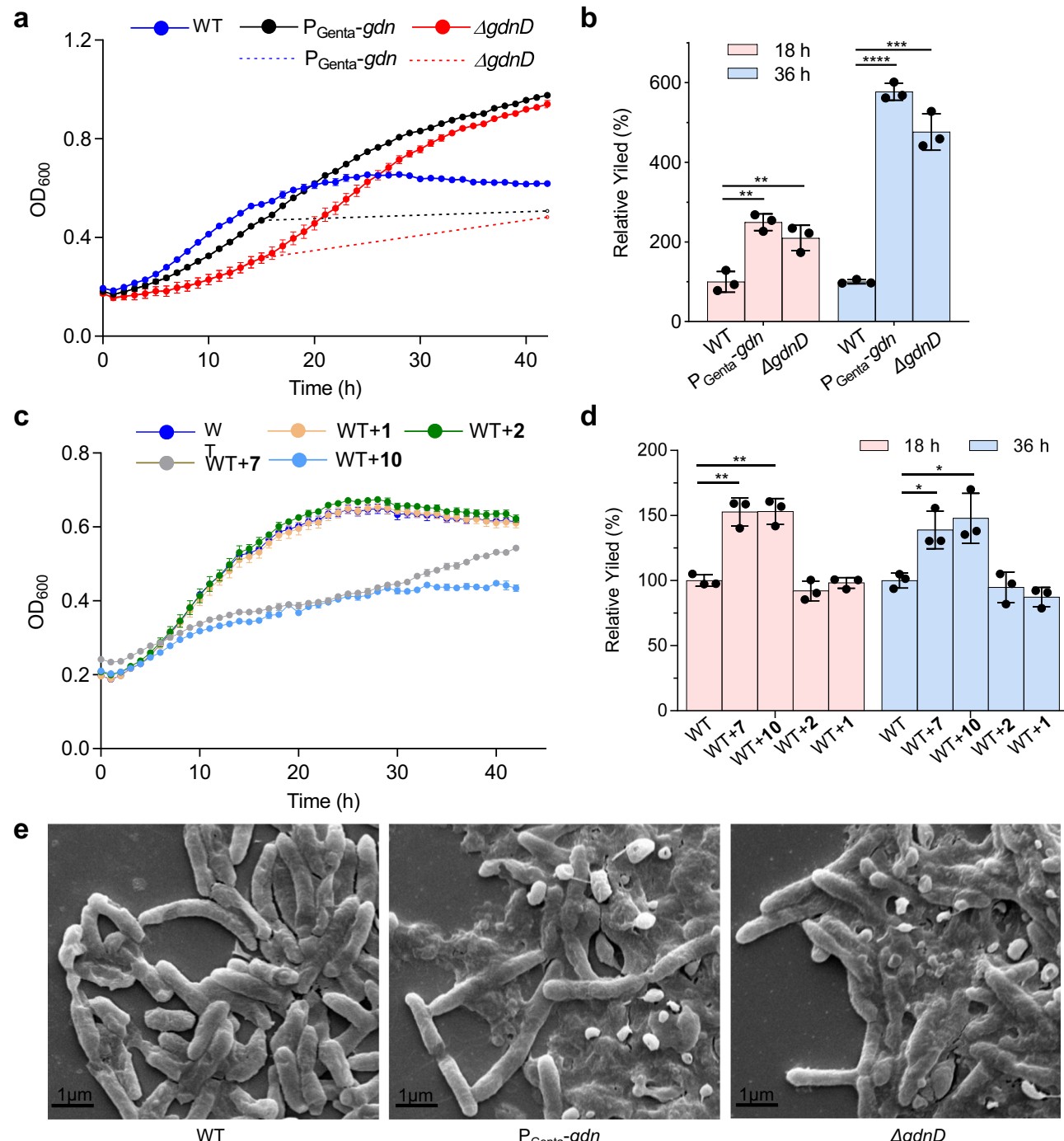

**Fig. 6 | The growth and biofilm formation between the wild type (WT) and mutants. a** The growth profiles of DSM 7029 WT and two mutants were measured by culturing in CYMG medium. Data are presented as mean values ± SD, $n = 3$ biologically independent samples. The dotted line means the authentic growth curves which the pigment absorption was deducted. **b** The comparison of the biofilm forming capabilities between WT and two mutants. The yield of the biofilm formation of the wild type strain was quantified as a reference (100%). Data are presented as mean values ± SD, $n = 3$ biologically independent samples. **c** The growth profiles of DSM 7029 WT and wild-type strain dealt with different compounds were measured by culturing in CYMG medium. Data are presented as mean values ± SD, $n = 3$ biologically independent samples. **d** The comparison of the biofilm forming capabilities between WT and wild-type strain dealt with different compounds. The yield of the biofilm formation of the wild type strain treated with $H_2O$ was quantified as a reference (100%). Data are presented as mean values ± SD, $n = 3$ biologically independent samples. All $P$ values were determined by two-tailed unpaired $t$ test. *$p < 0.05$, **$p < 0.01$, ***$p < 0.001$, ****$p < 0.0001$. **e** Cell morphology determination by using SEM of WT and two mutants. Source data and the exact $p$ values are provided as a Source Data file.

with improved bioactivities by introducing a C-terminal Put tail. The structural characterization, biosynthetic machinery and bioengineering of lipopeptides with a C-terminal Put offer a versatile platform for the combinatorial biosynthesis of lipopeptides.

## Methods

### Bacterial strains and reagents
All strains, plasmids, and primers used in this study are supplied in Supplementary Data 1. *S. brevitalea* DSM 7029 and mutant strains were

cultured on CYMG (Casein peptone 8 g/L, Yeast extract 4 g/L, MgCl$_2$ 6H$_2$O 8.66 g/L, Glycerol 5 mL/L) broth or agar plates with kanamycin (km, 30 µg/mL), gentamicin (genta, 20 µg/mL) and apramycin (apra, 30 µg/mL) at 30 °C. *E. coli* cultures were grown in Luria-Bertani (LB) broth or on LB-agar plates (1.2% agar) with ampicillin (amp, 50 µg/mL or 100 µg/mL), km (10 µg/mL or 15 µg/mL), or chloramphenicol (cm, 10 µg/mL or 15 µg/mL). Regents were purchased from Sigma-Aldrich, New England Biolabs, Invitrogen, and Thermo Fisher Scientific. DNA primers were purchased from Sangon Biotech, Tsingke Biological Technology, and RuiBiotech. The synthesized peptides (from Sangon Biotech) were shown in Supplementary Table 14, and their structures were confirmed by HPLC-ESI-MS, $^1$H and $^{13}$C spectra, which were shown in supplementary figures.

### Gene replacement in the chromosome of the activation mutant *S. brevitalea* DSM 7029 P$_{Genta}$-*gdn*

Our previous study demonstrated the *gdn* gene cluster was silent, and only three antibiotics (kanamycin/apramycin/gentamicin) were used in the *S. brevitalea* DSM 7029, as well two antibiotics (apramycin and gentamicin) are overlapping[40,43]. Thus, we first constructed the activated mutant *S. brevitalea* DSM 7029 P$_{Genta}$-*gdn*. The polymerase chain reaction (PCR) product of the constructive P$_{Genta}$ promoter with 50 bp homology arm was amplified. The template for gentamicin resistance gene is pFox-genta-P$_{tet}$-LTX. The PCR product was transformed into *S. brevitalea* DSM 7029 containing recombineering system Redγ-Redαβ7029. Correct recombinants were verified by colony PCR and DNA sequencing.

The single genes (*gdnC*, +1, −1, −2, −3, three PvdQ-like acylase genes, two formyltransferase PurH and PurN genes) were replaced by apramycin resistance gene (*apra$^R$*) using the recombineering system Redγ-Redαβ7029 as well, respectively. The apramycin resistance genes flanked with homology arms (50 bp) were generated by PCR amplification using 2 × Apex*HF* HS DNA Polymerase FS Master Mix, and the template for *apra$^R$* is plasmid RK2-apra-cm. Purified PCR products of apramycin resistance genes were transformed into *S. brevitalea* DSM 7029P$_{Genta}$-*gdn*/pBBR1-Rha-Redγ-Redαβ7029-km, respectively. Correct recombinants were verified by colony PCR.

### Site-directed mutagenesis and knockout of starter condensation (Cs) domain and module 13 on the *gdn* BGC

Due the site mutation of the *gdn* BGC was unachieved in DSM 7029, the *gdn* BGC was firstly cloned from the genome via direct cloning mediated by ExoCET[66] and verified by digestion by the restriction enzyme *ApaL*I. Then, the constructive P$_{Apra}$ promoter and *amp-attP* fragment were inserted into the plasmid p15A-cm-*gdn* by the linear plus circle homology recombination (LCHR) mediated by the recombineering system Redγ-Redαβ, respectively, to drive the whole *gdn* BGC and complement *gdn* gene cluster into DSM 7029. The correct plasmid p15A-amp-attP-P$_{Apra}$-*gdn* was confirmed by restriction enzyme and DNA sequencing.

We introduced several modifications into the plasmid p15A-amp-attP-P$_{Apra}$-*gdn* using the RedEx strategy[67], including knockout of the Cs domain and module 13, deletion of TE$_2$ domain (271 amino acids), TE domain (502 amino acids), and C$_{13}$A*T$_{13}$TE (1237 amino acids), specific mutations of Cs and module 13 at various sites (including H9721A in C domain, S10255A in T domain, and ΔTE$_1$-S10391A, ΔTE$_2$-S10624A, and ΔTE$_{1/2}$-S10391A/S10624A in TE domain), as well as the insertion of promoter P$_{13}$ in the front of TE domain (Supplementary Fig. 4a). The *BstZ17*I-cm-ccdB-*BstZ17*I cassette containing mutation site with 40 bp homology arm by PCR amplification can be stitched with the insertion sequence using overlap extension PCR. Two *BstZ17*I sites were flanked by 30-bp direct repeats. The *BstZ17*I-cm-ccdB-*BstZ17*I cassette was inserted into the target site by LCHR in the *E. coli* strain GBRed-gyr462. Correct recombinant plasmids were selected by restriction digest analysis. The *cm-ccdB* cassette was removed by *BstZ17*I digestion and

the terminal 30-bp overlaps were exposed. The linear fragments were circled by T4pol reaction in vitro and then the mixture was transformed into *E. coli* GB05. The resulting recombinant constructs were analyzed by restriction analysis and confirmed by DNA sequencing.

The fragments containing mutations and knockout of Cs domain were obtained by PCR using primer gdnCs-150HA-primer-S/gdnCs-150HA-primer-A, respectively. Then, the fragments were transferred into the mutant *S. brevitalea* DSM 7029ΔCs-genta containing recombineering system Redγ-Redαβ7029, which was constructed by the *genta* fragment replacing the Cs domain, generating two mutants of Cs domain. The region (~49.5 kb) containing *gdn* gene cluster was replaced in-situ by the *attB-genta* fragment in the strain *S. brevitalea* DSM 7029 containing the recombineering system Redγ-Redαβ7029, and we obtained a recombinant *S. brevitalea* DSM 7029:attB-genta (Supplementary Fig. 4c, d). In addition, the above nine correct mutant plasmids containing different deletions and mutations of module 13, the promoter P$_{13}$ insertion[53], and wild type plasmid p15A-amp-attP-P$_{Apra}$-*gdn* were transformed into the recombinant *S. brevitalea* DSM 7029:attB-genta generating different recombinants. The recombinants were verified by PCR and DNA sequencing. All plasmids generated in this study and oligonucleotides are listed in Supplementary Data 1.

### Multi-domain swapping of C-terminus of *gdn* BGC

The different kind of multi-domain swapping was conducted by LCHR and linear plus linear homologous recombination (LLHR). For engineering *rzmA** gene cluster, a linear fragment *cm-ccdB* cassette flanked with homology arms was amplified from plasmid R6K-cm-ccdB by PCR. Then, the *cm-ccdB* cassette and plasmid p15A-apra-phiC31-P$_{km}$-rzmA* (p15A-apra-phiC31-P$_{km}$-rzmCsR148A) were transformed into *E. coli* GBRed-gyr462 and recombined to form p15A-apra-phiC31-P$_{km}$-rzmA*-cm-ccdB. Another linear fragment T$_{12}$C$_{13}$A*T$_{13}$TE amplified from *gdn* gene cluster recombined with a linear p15A-apra-phiC31-P$_{km}$-rzmA*-cm-ccdB digested by *Pme*I yielding target plasmid p15A-apra-phiC31-P$_{km}$-rzmA*-M1 containing T$_{12}$C$_{13}$A*T$_{13}$TE domains of *gdn*. For another construct of *rzmA** gene cluster, another fragment *cm-ccdB* cassette with homology arms recombined with p15A-apra-phiC31-P$_{km}$-rzmA*-M1 to obtain p15A-apra-phiC31-P$_{km}$-rzmA*-M1-cm-ccdB, which was linearized by digestion of *Pme*I in the next. The linear T domain of *rzmA* by PCR recombined with the above linear *rzmA*-M1-cm-ccdB* to form plasmid p15A-apra-phiC31-P$_{km}$-rzmA*-M2 containing C$_{13}$A*T$_{13}$TE domains of *gdn* BGC.

For constructing the recombinants of the *holA* BGC, the procedure was the similar to the mutants of *rzmA**. The linear fragment *cm-ccdB* cassettes flanked with homology arms were amplified from R6K-cm-ccdB by PCR using different primers. The acquired *cm-ccdB* fragments were recombined to p15A-apra-phiC31-P$_{km}$-holA in *E. coli* GBRed-gyr462 by LCHR, respectively, which yielded different recombinant plasmids containing *cm-ccdB*. Then, fragments T$_{12}$C$_{13}$A*T$_{13}$TE, T$_{12}$C$_{13}$A*T$_{13}$, C$_{13}$A*T$_{13}$TE, T$_{12}$C$_{13}$A*, A*T$_{13}$TE, C$_{13}$A*, and C$_{13}$ were amplified from *gdn* gene cluster, respectively. The different plasmids containing *cm-ccdB* were linearized by digestion of *Pme*I. The PCR products and corresponding linear plasmids recombined to form target recombinant plasmids p15A-apra-phiC31-P$_{km}$-holA-M1, p15A-apra-phiC31-P$_{km}$-holA-M6, p15A-apra-phiC31-P$_{km}$-holA-M3, p15A-apra-phiC31-P$_{km}$-holA-M9, p15A-apra-phiC31-P$_{km}$-holA-M4, p15A-apra-phiC31-P$_{km}$-holA-M5, and p15A-apra-phiC31-P$_{km}$-holA-M8. In addition, two fragments T$_5$ and T$_6$ were amplified from *holA* BGC by PCR, which were fused with *cm* cassette, respectively, obtaining two linear products T$_5$-cm and T$_6$-cm. The fragment T$_5$-cm recombined with p15A-apra-phiC31-P$_{km}$-holA-M1 in *E. coli* GB08Red. Likewise, T$_6$-cm recombined with p15A-apra-phiC31-P$_{km}$-holA-M6 in *E. coli* GB08Red. Subsequently, the final plasmids p15A-apra-phiC31-P$_{km}$-holA-M2 and p15A-apra-phiC31-P$_{km}$-holA-M7 were obtained from the two recombinant plasmids by RedEx[67]. The melting sites were shown in Supplementary Fig. 25.

## Metabolic analysis

The fermentation and extraction of metabolites from the activation mutant and other mutants were the same as mentioned in our previous study[40]. The crude extracts were analyzed by UHPLC-HR-MS and LC–MS, respectively. The UHPLC system was performed using an ODS column (Luna RP-$C_{18}$, 4.6 × 250 mm, 5 μm, 0.75 mL/min) with gradient elution. Mass spectra was acquired in centroid mode ranging from 100 to 1500 $m/z$ with positive-mode electrospray ionization and auto $MS^2$ fragmentation. HPLC parameters were as follows: solvent A, $H_2O$ with 0.1% formic acid (FA); solvent B, 0.1% FA in acetonitrile (ACN); gradient at a constant flow rate of 0.75 mL/min, 0–5 min, 5% B; 5–55 min, 5–95% B; 55.1 min, 95% B; 55.1–60 min, 95% B; detection by UV spectroscopy at 200–400 nm. The HPLC–MS or LC–MS system was performed using Acclaim™ RSLC 120 $C_{18}$ column (2.1 × 100 mm, 2.2 μm) and its condition was: 0–3 min, 5% ACN; 3–18 min, 5–95% ACN; 18.1 min, 95% ACN; 18.1–22 min, 95% ACN; 22.1 min, 5% ACN; 22.1–25 min, 5% ACN, ACN and $H_2O$ contained 0.1% FA; flow rate by 0.3 mL/min; detection by UV spectroscopy at 200–400 nm.

## Purification of glidonins and lipopeptide derivatives

For compounds produced by *gdn* BGC of *S. brevitalea* DSM 7029 $P_{Apra}$-*gdn*, gradient elution ($CH_2Cl_2$: MeOH) resulted in 40 fractions. Fr.34-Fr.36 ($CH_2Cl_2$: MeOH 1:1) were purified by preparative MPLC (ODS; 5 μm, 250 × 20 mm, gradient elution 0–5 min, 20% MeOH; 5–55 min, 20-70% MeOH; 55.1 min, 95% MeOH; 55.1 min–60 min, 95% MeOH; $H_2O$ with 0.1% trifluoroacetic acid (TFA), 8 mL/min) to afford to Fr.1-Fr.4. Fr.2 was further purified by semipreparative HPLC (ODS; 5 μm, 250 × 10 mm, gradient elution 0–5 min, 30% ACN; 5–35 min, 30–60% ACN; 35.1 min, 95% ACN; 35.1–40 min, 95% ACN; $H_2O$ with 0.2% TFA, 3 mL/min) to yield compound **1** (100 mg). Fr.28-Fr.29 ($CH_2Cl_2$: MeOH 1:1) were purified by preparative MPLC (ODS; 5 μm, 250 × 20 mm, gradient elution 0–5 min, 10% MeOH; 5–55 min, 10-70% MeOH; 55.1 min, 95% MeOH; 55.1 min–60 min, 95% MeOH; $H_2O$ with 0.1% TFA, 8 mL/min) to afford to Fr.1-Fr.7. Fr.5 was further purified by semipreparative HPLC (ODS; 5 μm, 250 × 10 mm, isogradient elution 30% ACN, $H_2O$ with 0.2% TFA, 3 mL/min) to yield compound **2** (30 mg), Fr.7 was further purified by semipreparative HPLC (ODS; 5 μm, 250 × 10 mm, gradient elution 0–5 min, 34% ACN; 5–25 min, 34-48% ACN; 25.1 min, 95% ACN; 25.1 min–30 min, 95% ACN; $H_2O$ with 0.2% TFA, 3 mL/min) to yield compound **4** (28 mg). Fr.20-Fr.24 ($CH_2Cl_2$: MeOH 3:1) were isolated by negative column HPLC (MeOH: $H_2O$) and resulted in 10 fractions. Fr.8 (MeOH: $H_2O$, 60%) was further purified by semipreparative HPLC (ODS; 5 μm, 250 × 10 mm, gradient elution 0–5 min, 55% ACN; 5–38 min, 55-85% ACN; 38.1 min, 95% ACN; 38.1–43 min, 95% ACN; $H_2O$ with 0.2% TFA, 3 mL/min) to yield compound **7** (14 mg), compound **9** (20 mg), and compound **10** (15 mg). Compound **6** (7 mg) and **8** (5 mg) were purified from Fr.7 by semipreparative HPLC with the condition: 0–5 min, 55% ACN, 5–23 min, 55–65% ACN, 23.1 min, 95% ACN, 23.1–28 min 95% ACN; $H_2O$ with 0.2% TFA, 3 mL/min.

Glidonin A (**1**): colorless oil; $[\alpha]^{20}$D-25 (*c* 0.1, MeOH); UV (MeOH) $\lambda_{max}$ 192 nm; IR (KBr) $v_{max}$ 3308, 2969, 1868, 1681, 1540, 1455, 1207, 1138, 841, 802, 724 $cm^{-1}$; $^1H$ and $^{13}C$ NMR, Supplementary Table 2; HR-ESI-MS: $m/z$ 1359.7254 $[M+H]^+$ (calculated for $C_{65}H_{99}N_{16}O_{14}S$, 1359.7242). Glidonin B (**2**): colorless oil; $[\alpha]^{20}$D-27 (*c* 0.17, MeOH); UV (MeOH) $\lambda_{max}$ 191 nm; IR (KBr) $v_{max}$ 3311, 2969, 1868, 1681, 1541, 1454, 1204, 1137, 838, 802, 723 $cm^{-1}$; $^1H$ and $^{13}C$ NMR, Supplementary Table 3; HR-ESI-MS: $m/z$ 1375.7206 $[M+H]^+$ (calculated for $C_{65}H_{99}N_{16}O_{15}S$, 1375.7191). Glidonin D (**4**): colorless oil; $[\alpha]^{20}$D-20 (*c* 0.12, MeOH); UV (MeOH) $\lambda_{max}$ 191 nm; IR (KBr) $v_{max}$ 3309, 2969, 1868, 1653, 1540, 1455, 1205, 1138, 839, 802, 723 $cm^{-1}$; $^1H$ and $^{13}C$ NMR, Supplementary Table 4; HR-ESI-MS: $m/z$ 1385.7381 $[M+H]^+$ (calculated for $C_{66}H_{99}N_{16}O_{15}S$, 1385.7046). Glidonin F (**6**): colorless oil; $[\alpha]^{20}$D-22 (*c* 0.1, MeOH); UV (MeOH) $\lambda_{max}$ 221 nm; IR (KBr) $v_{max}$ 3305, 2928, 1795, 1652, 1540, 1455, 1204, 1136, 836, 801, 723 $cm^{-1}$; $^1H$ and $^{13}C$ NMR, Supplementary Table 4;

HR-ESI-MS: $m/z$ 757.4335 $[M+2H]^{2+}$ (calculated for $C_{75}H_{117}N_{16}O_{15}S$, 1513.8600). Glidonin G (**7**): colorless oil; $[\alpha]^{20}$D-23 (*c* 0.12, MeOH); UV (MeOH) $\lambda_{max}$ 221 nm; IR (KBr) $v_{max}$ 3306, 2929, 1868, 1651, 1540, 1455, 1204, 1137, 837, 802, 722 $cm^{-1}$; $^1H$ and $^{13}C$ NMR, Supplementary Table 5; HR-ESI-MS: $m/z$ 771.4479 $[M+2H]^{2+}$ (calculated for $C_{77}H_{121}N_{16}O_{15}S$, 1541.8913). Glidonin H (**8**): colorless oil; $[\alpha]^{20}$D-28 (*c* 0.1, MeOH); UV (MeOH) $\lambda_{max}$ 221 nm; IR (KBr) $v_{max}$ 3310, 2969, 1868, 1681, 1541, 1454, 1204, 1137, 838, 802, 723 $cm^{-1}$; $^1H$ and $^{13}C$ NMR, Supplementary Table 5; HR-ESI-MS: $m/z$ 779.4772 $[M+2H]^{2+}$ (calculated for $C_{77}H_{121}N_{16}O_{16}S$, 1557.8862). Glidonin I (**9**): colorless oil, $[\alpha]^{20}$D-25 (*c* 0.15, MeOH); UV (MeOH) $\lambda_{max}$ 221 nm; IR (KBr) $v_{max}$ 3310, 2928, 1795, 1652, 1540, 1455, 1204, 1137, 837, 802, 723 $cm^{-1}$; $^1H$ and $^{13}C$ NMR, Supplementary Table 6; HR-ESI-MS: $m/z$ 762.4718 $[M+2H]^{2+}$ (calculated for $C_{78}H_{123}N_{16}O_{15}$, 1523.9348). Glidonin J (**10**): colorless oil; $[\alpha]^{20}$D-21 (*c* 0.18, MeOH); UV (MeOH) $\lambda_{max}$ 222 nm; IR (KBr) $v_{max}$ 3305, 2927, 1868, 1669, 1540, 1454, 1204, 1136, 836, 802, 722 $cm^{-1}$; $^1H$ and $^{13}C$ NMR, Supplementary Table 6; HR-ESI-MS: $m/z$ 785.4645 $[M+2H]^{2+}$ (calculated for $C_{79}H_{123}N_{16}O_{15}S$, 1569.9226).

To obtain the two derivatives **29** and **31**, a 16 L scale fermentation of the mutant *E. coli* GB05-MtaA:RzmA*-M2 and a 12 L scale fermentation of the mutant *E. coli* GB05-MtaA:HolA-M1 were prepared. The crude extracts of bacterial fermentation were purified by normal phase column and semipreparative HPLC. The condition of the gradient elution for generating **29** (3.6 mg) was 0–5 min, 40% ACN; 5–30 min, 40–65% ACN; 30.1 min, 95% ACN; 30.1–35 min, 95% ACN; $H_2O$ with 0.1% TFA, 3 mL/min. And the condition of the gradient elution for generating **31** (3.6 mg) was 0–5 min, 46% ACN; 5–24 min, 46–58% ACN; 24.1 min, 95% ACN; 24.1–29 min, 95% ACN; $H_2O$ with 0.1% TFA, 3 mL/min. In addition, the derivatives and intermediates (or original products) produced by the engineering *rzmA** and *holA* BGCs were quantified using the standard curve of compounds **28** (for quantification of compound **28**), **29a** (for quantification of compound **28a**), **29** (for quantification of compound **29**), **30a** (for quantification of compounds **30, 30a, 30b**), and **31** (for quantification of compound **31**) using HPLC–MS measurement as described above (5%–95% ACN in 15 min). Triplicates of all experiments were measured.

The mutant strain *E. coli* GB05-MtaA:RzmA*-M2 was cultured with three different diamines 1,7-diaminoheptane, 1,6-diaminohexane, and 1,5-diaminopentane with the concentration of 1.6 μg/mL, respectively. After three days, the XAD-16 resin was added into the fermentation, and the mixture was cultured continually for one day. Then, the fermentations were extracted using MeOH as previous study[40]. The crude extracts were analyzed by UHPLC–MS (5%–95% ACN in 50 min) as the above procedure.

## Marfey's analysis of the constitutes of glidonins

Approximate 0.5–1 mg of compounds was hydrolyzed with 6 M HCl (0.5 mL) at 90 °C overnight. The solutions were dried by nitrogen, and the mixture was dissolved in 200 μL of $H_2O$. The standard amino acids were also dissolved in 200 μL of $H_2O$. Then, the above samples were added 1 M $NaHCO_3$ (25 μL) and 1% 1-fluoro-2,4-dinitrophenyl-5-L-alanine amide (L-FDAA, 200 μL)[49], and the mixture was vortexed and incubated at 40 °C for 60 min. Then, 100 μL of 2 M HCl was added to quench the reaction and the filtered solution was analyzed by LC–MS using Acclaim™ RSLC 120 $C_{18}$ column (2.1 × 100 mm, 2.2 μm) with a linear gradient of ACN with elution conditions (5%–50% ACN in 35 min) at a flow rate of 0.3 mL/min and UV detection at 330 nm. The LC–MS condition: 0–3 min, 5% ACN; 3–38 min, 5–50% ACN; 38.1 min, 95% ACN; 38.1–42 min, 95% ACN; 42.1 min, 5% ACN; and 42.1–45 min, 5% ACN, ACN, and $H_2O$ contained 0.1% FA.

## Purification and characteristics of enzymes

Primers used for amplification of acylase GdnD, methionyl-tRNA formyltransferase (Fmt) from the genomic DNA of *S. brevitalea* DSM 7029, primers used for amplification of CsA₁T₁, A₁T₁, A₁₂T₁₂, A₁₂T₁₂C₁₃,

$A_{12}T_{12}C_{13}A^*$, $A_{12}T_{12}C_{13}A^*T_{13}$, $A_{12}T_{12}C_{13}A^*T_{13}TE$ from the plasmid p15A-cm-*gdn* are listed in Supplementary Data 1. The mutated fragments including $A_{12}T_{12}C_{13}A^*T_{13}TE_{Mut1}$(S10391A), $A_{12}T_{12}C_{13}A^*T_{13}TE_{Mut2}$ (S10624A), and $A_{12}T_{12}C_{13}A^*T_{13}TE_{Mut1/2}$(S10391A/S10624A), and others were obtained by fused PCR. The resulting DNA fragments were cloned into the commercial pET-28b(+) vector via LLHR were transformed into *E. coli* GB05Dir. Then, the resulting constructs were verified by restriction digestion and DNA sequencing. The correct plasmids were transformed into *E. coli* BL21 or *E. coli* BAP1 for expressing the His-tagged protein. Each recombinant *E. coli* BL21 or *E. coli* BAP1 was incubated in LB medium supplemented with km (10 µg/mL) for overnight at 37 °C. Overnight 30 mL culture was transformed into 2 L fresh LB medium supplemented with km at 37 °C, 180 rpm for 2 h, and the growth was continued to an $OD_{600}$ reached to 0.6 at 16 °C, 160 rpm. Then, the addition of 0.2 mM isopropyl-β-D-thiogalactopyranoside (IPTG) to the cultures that were further incubated at 16 °C for additional 20 h. For protein purification according to our previous study[41], the cells were harvested via centrifugation (8,228 × g, 4 °C, 10 min). The pellet was responded in buffer A (50 mM Tris, 300 mM NaCl, 5 mM $MgCl_2$, 40 mM imidazole, pH 7.6), and lysed by cell disruption. Cell debris was removed by centrifugation (60,325 × g, 4 °C, 30 min), and the supernatant was bound to Ni-NTA resin. The protein was eluted with buffer B (50 mM Tris, 300 mM NaCl, 5 mM $MgCl_2$, 400 mM imidazole, pH 7.6). The buffer of the elution fraction was exchanged with PD10 column using buffer C (50 mM Tris, 300 mM NaCl, 5 mM $MgCl_2$, pH 7.6). Then, proteins were concentrated with centrifugal filters with a molecular weight cut off (30 kDa). Finally, proteins were portioned into 100 µL aliquots, flash frozen on liquid $N_2$, and stored at −80 °C.

For pyrophosphate production assays[68], the reaction system contains 10 mM amino acid substrates, 2.5 mM ATP (Solarbio), 0.2 mM MESG (Medchemexpress), 1 U/mL purine nucleoside phosphorylase (Shanghai yuanye Bio-Technology Co., Ltd), 0.4 U/mL inorganic pyrophosphatase (Sigma), 150 mM hydroxylamine (Sigma) and 1 µM proteins in 100 µL buffer (50 mM Tris, 300 mM NaCl and 5 mM $MgCl_2$, pH 7.6). The value was read at A360 in a SpectraMax M5 plate reader (Molecular Devices, Sunnyvale, California) in 96 well, clear-bottomed plates (Corning). A360 values were converted to pyrophosphate release, by comparing with a standard curve ($KH_2PO_4$ from 2 µM to 128 µM) for known quantities of pyrophosphate.

In vitro assay for $CsA_1T_1$ domain was performed. Reaction buffer (50 mM Tris, 300 mM NaCl and 5 mM $MgCl_2$, pH 7.6) contained 80 µM protein, 3 µM *Bacillus subtilis* phosphopantetheinyl transferase (Sfp), 200 µM Coenzyme A (CoA), 2.5 mM substrates of amino acids (i.e. L-Met, L-Met(O) and L-Leu respectively), and 5 mM ATP was incubated at 30 °C for 30 min. Then, 5 mM acyl-CoA (C12-CoA and C14-CoA respectively) was added to make final volume of 200 µL. The reaction was further incubated at 30 °C for 4 h. Products was hydrolyzed as reported[7]. In brief, triple amount of cold methanol was added to quench the reaction, and the precipitate was collected at 10,000 × g for 8 min. The pellet dissolved in 125 µL of 0.1 M KOH and was heated at 70 °C for 10 min after washed twice and dried. Then the protein was neutralized with HCl after cooling to room temperature. One milliliter of the methanol was added to precipitate the protein overnight at −20 °C. The hydrolysate was centrifuged at 10,000 × g for 15 min to remove precipitated protein, and the supernatant was applied LC–MS analysis (5%–95% ACN in 15 min).

Protein-MS of $T_1$ domain linked substrates was conducted as the previous study[69]. Each sample (100 µL) was completely mixed with 400 µL methanol, 100 µL chloroform, and 300 µL $ddH_2O$ by gentle vortexing. Samples were centrifuged for 15 min at 18,407 × g, 4 °C. The top supernatant of each sample was aspirated carefully to not disturb thin layer of the precipitated protein located within the organic/aqueous interface. An additional 400 µL of methanol was added to each sample again followed by gentle vortexing, and the mixture centrifuged for 15 min. Solvent was completely removed and the pellet was

dried briefly. To ensure complete removal of interfering substances, an additional 400 µL of methanol was added to each pellet. The resulting pellets were chilled at −20 °C for 15 min and incubated in 25 µL of 80% FA at −20 °C for 2 min. The samples were then mixed and incubated at −20 °C for an additional 15 min. 75 µL of cold $ddH_2O$ was then added to each sample, followed by a final 1:50 dilution in 49.95% ACN, 49.95% $ddH_2O$, and 0.1% FA (v/v). In all, 4 µL sample was detected using a UHPLC system coupled to an impact HD QTOF (Bruker) with ACQUITY UPLC Protein BEH $C_4$ column (300 Å, 1.7 µm, 2.1 mm × 100 mm). For the LC, Solvent A: 95% $H_2O$, 5% ACN, 0.2% FA and Solvent B: 5% $H_2O$, 95% ACN, 0.2% FA were used with the following gradient: 0–5 min, 5% B; 5–42 min, 5–60% B; 42–44 min, 60–95% B; 44–46 min, 95% B; 46–47 min, 95-5% B; 47–60 min, 5% B, at a flow rate of 0.2 mL/min with the mass range at 600–2000 m/z.

In vitro assays for the different enzymes of the C-terminal multi-domains were performed. Reaction buffer contained 80 µM protein, 3 µM Sfp from *B. subtilis*, 200 µM CoA, 2.5 mM L-Ala and 5 mM ATP was incubated at 30 °C for 30 min. Then, 2.5 mM diamines or polyamines (i.e. putrescine, 1,7-diaminoheptane, 1,6-diaminohexane, 1,5-diaminopentane, 1,2-diaminocyclohexane, spermine, and spermidine, respectively) was added to make final volume of 200 µL. The reaction was further incubated at 30 °C for 1 h or 4 h. Triple amount of cold methanol was added to quench the reaction, and the supernatant was dried by SpeedVac after removed the precipitate at 10,000 × g for 8 min. The reaction mixture was dissolved with 200 µL $ddH_2O$, and then the mixture was derived with D-FDAA according to the procedure of the above Marfey's analysis. The final reaction mixture was suspended using 100 µL 2 M HCl, which subsequently was dried by SpeedVac. Finally, the mixture dissolved with 100 µL MeOH to detect the target products (**24a, 25a, 26a**, and **27a**) using LC–MS analysis (5%–95% ACN in 15 min).

The hydrolysis function of PvdQ-like acylase GdnD was tested. GdnD is predicted to be a periplasmic protein by PSORTb[13]. The collected pellet from 50 mL LB dissolved with 8 mL buffer C (50 mM Tris, 300 mM NaCl, 5 mM $MgCl_2$, pH 7.6), and lysed by cell disruption. Then, one hundred microliter of the cell lysate was incubated with 5 µL 10 mM substrates **7** and **8** for 4 h, 16 h, and 27 h at 30 °C. The negative control was boiled cell lysate. The other substrates (**13, 14, 15, 16**, and **17**) incubated with the cell lysate dealt for 16 h at 30 °C. Then, the reaction was stopped by the addition of the 100 µL MeOH, and the final mixture was checked by LC–MS analysis (5%–95% ACN in 15 min) according to the above condition.

The in vitro assay of formyltransferase was tested[70]. In brief, the formyl-donor co-substrate intermediate $N^5$, $N^{10}$-methenyl$H_4$F was first synthesized as follow: 14 mg of $N^5$-f$H_4$F was dissolved in 3 mL of water and was converted to $N^5$, $N^{10}$-methenylH⁴F through addition of 0.1 M HCl, until pH 1.9 was reached. The solution was then brought to a final volume of 4.4 mL with water and continually incubated at room temperature for 4 h. Then, the transformylation assay was setup in a 50 µL volume, in the presence of 50 mM HEPES buffer pH 7.5, 1 mM substrates (compounds **1, 1a, 13a**), 1.5 mM $N^5$, $N^{10}$-methenyl$H_4$F, and 25 µM Fmt (35 kDa). Before the addition of the compounds and the enzyme, the intermediate $N^5$, $N^{10}$-methenyl$H_4$F was preincubated in the reaction buffer at 30 °C for 30 min, to permit the final pH-dependent conversion to $N^{10}$-f$H_4$F. After the addition of the compounds and the enzyme, the reaction was allowed to proceed for 4 h at 30 °C, then ceased by the addition of 2 µL FA, and analyzed via HPLC–MS. The condition for **1**: 0–3 min, 20% ACN; 3–18 min, 20–60% ACN; 18.1 min, 95% ACN; 18.1–22 min, 95% ACN; 22.1 min, 20% ACN; 22.1–25 min, 20% ACN, ACN and $H_2O$ contained 0.1% FA; detection by UV spectroscopy at 200–400 nm. The condition for **1a** and **13a**: 0–3 min, 5% ACN; 3–18 min, 5–95% ACN; 18.1 min, 95% ACN; 18.1–22 min, 95% ACN; 22.1 min, 5% ACN; 22.1–25 min, 5% ACN, ACN and $H_2O$ contained 0.1% FA; detection by UV spectroscopy at 200–400 nm.

## Bioactivity assays screening of glidonins and derivatives

The two human acute lymphoblastic leukemia cells K562 and Kasumi, two human breast adenocarcinoma cells MCF7 and MDA-MB-231, human hepatoma cell HepG-2, human lung adenocarcinoma cell A549, human gastric cancer cell SGC-7901, human colon cancer cell HCT-116, human ovarian cancer cell line SKOV3, or human lung epithelial normal cell 2B were obtained from Beyotime Biotechnology and utilized to test the cytotoxicity of compounds by MTT assays as previous study[71]. The anti-inflammatory effect of the compounds was detected by the influence on the increase in the level of nitric oxide caused by lipopolysaccharide (LPS) based on the classic Griess method. RAW 264.7 cells (Shanghai Institutes for Biological Science) were seeded in 96-well plates and incubated for 24 h. Then, cells were treated with compounds **29, 29a, 31**, and **31a** (0-80 µM) for 30 min, followed by co-treatment with LPS (10 mg/mL) for another 24 h. The Nitric Oxide assay kit with Griess reagents (Beyotime, Lot: S0021, Shanghai, China) was used to examine cellular supernatant nitrite accumulation, which represents cellular NO levels. The OD value of 540 nm absorbance was detected with a microplate reader. Their authentication in the laboratory was performed regularly based on morphology and gene/protein expression. The tested micro-organisms including Gram-negative bacteria *E. coli* ATCC 35218 and *Pseudomonas aeruginosa* ATCC 27853, and Gram-positive bacteria *Staphylococcus aureus* ATCC 29213 and *B. subtilis* ATCC 6633, and fungi *Candida albicans*, *Rhizoctonia solani* and *Aspergillus fumigatus* were treated with different concentrations of compounds (1 µM, 10 µM, 100 µM) using Bauer-Kirby disk diffusion method to evaluate the antimicrobial activities[72].

Comparing the concentration of compounds (**7, 10, 10a**) in extracellular and intracellular after the HepG2 cell line dealt with compounds. HepG2 cells were plated in six-well plates at a density of $1.5 \times 10^5$ cells/well and grew overnight. After cell treated with compounds (5 µg/mL) for 3 h, the compounds were extracted with MeOH for cells culture medium and cells separately as followed the reported method[73]. The compounds were quantified by LC–MS (5%–95% ACN in 15 min). The cell numbers of HepG2 with the compounds treatments was determined.

## Physiology and biochemistry test

The growth curve was tested by Bioscreen C (Labsystems, Finland) as follow. Overnight culture was inoculated into 400 µL fresh CYMG medium in the ratio of 1:100 (the concentration of compounds was 0.1 µg/µL) and shook in normal speed. The $OD_{600}$ value was gained in every hour for 42 h in total. Biofilm formation was measured at 18 h and 36 h respectively, as reported previously[74]. In brief, the free-floating cells were removed, and the microtitre plate was washed with PBS (pH 7.4) after incubation for 18 h and 36 h. The biofilm was stained by adding 200 µL of crystal violet (0.1%) solution. Crystal violet solution was removed after 15 min and 200 µL of ethanol (95%) was added. The absorbance at $OD_{600}$ was read using a microplate reader (Infinite M200, Tecan, Männedorf, Switzerland). Meanwhile, free-floating cells were used to test the secretion of pigment. The bacteria were removed and the supernatant was measured at $OD_{600}$.

## Rigid receptor docking

The structure of GdnD was predicted by AlphaFold2[54]. The structure of the PvdQ in complex with the pyoverdine precursor PVDIq (PDB number: 5UBK) was retrieved from RCSB Protein Data Bank (PDB). GdnD was docked with substrate C14-D-Met(O) in the PVDIq binding pocket of PvdQ using AutoDock Vina. To cover the entire binding site, a 2 nm side cubic grid box was positioned at the geometric center of residues S486 (PvdQ S268), N497 (PvdQ D269), R516 (PvdQ R297), and H683 (PvdQ H457) as reported[75]. A total of 20 poses for one substrate were generated through the docking procedure, and the complex that closely matched the binding pose of PVDIq in PvdQ was selected independently for further analysis.

The docking procedure for the $C_{13}$ domain with Put followed a similar approach. RzmA mutant H140A/R148A in complex with C8-CoA and Leu-SNAC (PDB number: 7C1S) was used as the model complex and 2 nm side cubic grid box was positioned at the geometric center of residues H9721 (RzmA H140), T9879 (RzmA T289), V9877 (RzmA S287), and L9860 (RzmA P271) as reported[56]. The complex that closely matched the binding pose of Leu-SNAC in RzmA and 4′-phospho-pantetheine in the condensation domain of holo-AB3403 (PDB number: 4ZXI[76]) was selected for further analysis.

## Reporting summary

Further information on research design is available in the Nature Portfolio Reporting Summary linked to this article.

## Data availability

Data supporting the findings of this work are available within the paper and its Supplementary Information files. A reporting summary for this Article is available as a Supplementary Information file. The following public databases were utilized for the analysis of the protein structures: 5UBK, 7C1S, 4ZXI, 3L91, 1JMK. Source data are provided with this paper.

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

## Acknowledgements

We thank Haiyan Sui, Xiangmei Ren, Zhifeng Li, Jingyao Qu, Jing Zhu, and Guannan Lin from State Key Laboratory of Microbial Technology of Shandong University for help and guidance in NMR, LC and HRMS. This work was supported by National Key R&D Program of China (2019YFA0905700), National Natural Science Foundation of China (32070060, 32000040, 32100052, 32371488, 32161133013, 32201195), Shandong Provincial Natural Science Foundation (ZR2020QH345), the 111 project (B16030), and Youth Interdisciplinary Innovative Research Group (2020QNQT009) and Fundamental Research Funds (2023QNTD001) of Shandong University.

## Author contributions

X.Bi. designed the research; X.Bi. and Yo.Z. supervised the experiments; H.C., L.Z., T.S., and X.Ba. performed gene cloning, site mutations, protein purification, in vivo deletion; H.C. and L.Z. performed bioinformatic analyses, in vitro experiments; H.C., X.W., and X.J. purified compounds and analyzed data; H.Z. performed structural elucidation; L.Z. performed structural prediction of proteins; Yi.Z. performed bioactivities of compounds; H.C., L.Z., H.Z., Q.T., and X.Bi. wrote and revised the paper.

## Competing interests

The authors declare no competing interests.
