## [Peer Review File · Nature Communications]

Biosynthesis and Engineering of the Nonribosomal Peptides with a C-terminal PutrescineREVIEWER COMMENTS

Reviewer #1 (Remarks to the Author):

The article by Chen et. al. reports the discovery, biosynthesis and engineering of a set of dodecapeptides – named glidonins – from *Schlegelella brevitalea*, that are notable for the presence of a putrescine appended to the C-terminus. This work appears to build upon the initial detection of glidonins in previous work (Proc. Natl. Acad. Sci. U S A., 115, E4255), for which these molecules have now been structurally elucidated and linked to a near-colinear NRPS.

A huge quantity of work is packed into this manuscript, which includes:

- 1). structural elucidation of glidonins
- 2). Biological testing of glidonins
- 3). identification of enzymes involved in deacetylation and formylation of glidonins – both of which are post-NRPS tailoring steps
- 4). substrate tolerance of the initiation module (A domain (MetOx) and C domain (fatty acyl chain)).
- 5). in vivo and in vitro characterisation of putrescine incorporation steps.
- 6). biological activity testing of glidonin +/- putrescine residue
- 7). domain swapping experiments to incorporate putrescine residues into other lipopeptides
- 8). substrate tolerance of C13 domain for other diamines
- 9). role of acylated form of glidonins in biofilm formation.

As a result, a general comment is that the manuscript is incredibly difficult to digest – with all of the above points discussed in heavy detail. I therefore suggest that sections 2, 3, 6 and 9 are reduced to keep the focus on the 'biosynthesis' and 'engineering' aspects of the paper, which are the take-home messages.

See below for specific points on the manuscript:

- 1). Ln99 – 'The products of BGC11 in DSM 7029...' – this is a very inaccessible way to start the results section. Consider revising this opening paragraph for readers who haven't read (Proc. Natl. Acad. Sci. U S A., 115, E4255).
- 2). Ln100 'Papa' – should be PApra
- 3). Ln113 - which are predicted to activate Phe, Asn, and Pro by A domain specificity analysis.' – this needs references.
- 4). HRESIMS – should be 'HR-ESI-MS'; throughout.
- 5). Ln157 & Sup Table 10 – please add chromatograms of Marfey's analysis to the SI to show retention time alignment with authentic standards.
- 6). Ln179 – 'The final products of Cs domain-embedded' – please define Cs domains.
- 7). Ln246 – the ATP/PPi exchange assay is good for assaying initial activation of the amino acid, but not the subsequent loading onto the T domain. Has this step been checked for MetOx and Met for the CsA1T1 protein? Perhaps using protein-MS? Also, was the CsA1T1 protein obtained from BAP1 cells completely in a holo- form for these assays?
- 8). The ability of the A1 domain to apparently selectively activate Met, MetOx and Leu is quite a strange set of substrate preferences. Have other amino acids been screened against the A1 domain? Some interesting amino acids to test would be: Ile, Val, Ala, Lys based on similarity to residues

incorporated into the final product.

9). Ln269/270 – TEMut1, TEMut2 and TEMut1/2 should be written as the amino acid mutations as in Fig 3b. Also, very confused as to which residues have been mutated – presumably one of the Ser residues is the active site (acylated) Ser of a canonical TE domain. What is the role of the other Ser residue? Does the TE domain not have an active site His? This whole section needs to be much clearer.

10). Figure 1 – a few points:

- add a physical space between the GdnA and GdnB subunits.
- add Ppant thiols to all T domains (maybe even add the loaded amino acid structures?)
- add lines over the top of module boundaries
- label putrescine

Reviewer #2 (Remarks to the Author):

The paper by Chen et al describes the identification of a new NRPS-derived peptide, glidonins, from *Schlegella brevitalea* DSM 7029.

Unusual features in this biosynthesis are the C-terminal attachment of a putrescine that was described in detail.

In general, I really like the paper, which presents a wealth of information, but I think the paper needs more clarifications at various positions to find the readers way through its content. The starter fatty acid diversity is nothing new and does not need to be presented that prominently. Putrescine as C-terminus was also described before and also the use of terminal amine incorporating domains was shown in detail before in other examples (see below). However, the authors nicely figured out what part of their NRPS was responsible for it.

Experimental details are missing at several parts, as well as structure information for some compounds that were made as controls. Please see my comments below:

- L. 63 ff: there are many more natural products containing C-terminal amines (phenylethylamine, spermidine, tryptamine etc) including putrescine: Cai X et al, Nat Chem 2017; Zhao L et al Org Lett 2018; Paenilamicin Angew Chem 2014, etc
- L 116: can the original specificity of the truncated A domain be suggested/postulated?
- L. 117: what is the distance between the two possible active sites?
- L. 168: lauroylation and lauroyl
- L. 203: How was the docking performed in all docking examples in the paper? Please add more details about modelling and docking to the methods or the SI.
- L 219 ff: the finding of a formylated peptide in *E. coli* is interesting but is this relevant in the original producer? Were formylated products also found there? If not, please remove this part from Fig. 1 since it might not be part of the pathway. If the formyl derivatives are found in the DSM strain, the authors might be able to perform an in vitro assay with the methionyl-tRNA formyltransferase as most likely candidate for the modification.
- L 228: Formylation in szentiamide: see Bode et al Angew Chem 2019 (SI)
- L 289: Why was 24 derivatized with D-FDAA and what is the structure of 24a? Where is this described in the methods?
- L 345: do the authors know if the compounds with and without putrescine address the same target or can they measure uptake into the cell? If the state about the improvement of the bioactivity they imply the same target of both compounds but it can also be different targets.
- L 357/374: what is the structure of 29 and 29a and 31 and 31a? How were they synthesized and confirmed? Why was NO production measured? Explain in more detail also in the methods.
- L 389 ff: the use of terminal amine incorporating domains to generate different amine derivatives was already shown extensively in Cai et al, Nat Chem 2017. What is the bioactivity of the non-natural derivatives?

- L 442 f: How does the biofilm modulation fits to the also described cytotoxic activity?
- Fig. 1: Explain in the text more about the truncated A domain or show an alignment in the SI so one can see what happened here?
- Fig. 2. In d-g: show both compounds, substrate and product in the chromatograms as in c. Change lauricyl to lauroyl. What is the substrate specificity regarding the amino acid composition? The authors have tested Met only but can include other amino acids as well. Where are the structures of 16 and 17? All numbers must be reflected by structures either in the text or the SI.
- Fig. 4: explain the docking method
- Fig. 6. Explain the use of D-FDAA in methods. Isolate and test the derivatives or move whole figure into the SI since the results are not too surprising but expected from previous work by others (see above).

SI:

- Give solvent and field for all NMR data in Tables and figures
- Fig S1. Give wt control as well
- S2c: the chromatogram looks rather different than that in S1. Please explain and clarify in methods.
- S3: Show all peptides linear from N-C from left to right (in all Figures also in S9 etc). Do not bend the molecules since it makes it more difficult to read. 9 is missing part of th aromatic ring.
- S8: ad domain organization, otherwise one can not understand the text on line 263.
- S16: give more details in the text/methods about the modelling and the performed analysis.
- NMR data: I do not find the labelling of the peaks and the integrals very useful in the spectra since there is a table with all this data. It would be nicer to assign these signals or leave the label out and refer to the tables only (but still show the spectra).

Reviewer #3 (Remarks to the Author):

This manuscript reports the discovery of glidonins and their biosynthesis and biological activities. A lot of data were presented. The role of the termination module in putrescine incorporation is perhaps most interesting. A C13 domain was suggested to catalyze the incorporation, and the TE domain was somehow also involved. This termination module was swapped into other nonribosomal peptide synthetases and used to introduce putrescine into the termini of two different peptides. The difference in bioactivity between the acylated and deacylated variants is also interesting. However, the precise role of the TE domain in putrescine incorporation was not determined, and the efficiency of the engineered pathways did not take incomplete or shunt products into consideration. Description of several experiments lacks key information, such as concentration of compound used. Overall, although new insights were made on a silent gene cluster, the manuscript is poorly written with many grammatical and spelling mistakes, it lacks statistical analysis, and the conclusions are diffused. Specific comments are included below:

1. Line 108-109, "while the other mutants fail to impact on the production of glidonin" is misleading. Looks like mutants of gdnA and gdnB were not made.
2. Line 117, sequence alignment that shows the two conserved active sites in the TE domain should be included in the SI.
3. Line 138, "rotamer" was not the correct term to use here.
4. Figure 1b, would be helpful to specify which ones are the dual C domains.
5. Line 177, the authors mentioned that glidonins had no antimicrobial activities, but later on in Figure 7c, 7 and 10 were found to inhibit the growth of the producing strain. This seems conflicting, and the names of the bacterial strains tested should be provided.

6. Line 270, Fig. 3e was referenced, but there is no panel e in Figure 3.
7. Are the same residues in PvfQ conserved in GdnD for substrate binding? What are the sequence similarity between the two proteins? Rationalization of the promiscuity of GdnD was vague.
8. No statistical significance was shown in Figure 4 or Figure 7 to compare the activities of the different mutants.
9. The three residues identified in C13 are in a loop region and unlikely to be accurate in the model. It is possible that the alanine mutations at this loop region affected the local folding and lost activity. Conclusion drawn from RMSD analysis is very speculative.
10. Figure 4c legend, need to keep the residue numbers consistent with the full length protein.
11. SI Figure 3, the phenylalanine ring was cut off in the structure of 9.
12. In domain swapping experiments, were incomplete or shunt products formed? If so, at what levels? What inter domain-linkers were used? What were the isolated yields of the desired compounds in Figure 5e,f,g? Were the bioactivity tested on isolated compounds? Such information would be important to evaluate the efficiency of the NRPS engineering experiments.
13. Figure 5h and 5i were barely discussed in the text with regard to the anti-inflammatory activities. The LPS-induced NO production was not explained at all.
14. Figure 7c and d, what concentrations were the compounds?

This paper has many spelling and grammatical mistakes. I've only listed a few examples:

1. Line 63-64, "amino acid moieties loaded in lipopeptides, which was proposed that this process might..." is awkward.
2. Throughout the manuscript, lauricyl should be lauroyl, myristicyl should be myristoyl.
3. Line 114, define X.
4. Line 119, should be "ten conserved active site residues"
5. Line 129, "revealed that 1 was confirmed to be" is awkward.
6. Line 205, "opened" should be "open"
7. Line 274, "to furtherly clarify"
8. Line 307-309, confusing sentence.

REVIEWER COMMENTS

Reviewer #1 (Remarks to the Author):

The article by Chen et. al. reports the discovery, biosynthesis and engineering of a set of dodecapeptides – named glidonins - from *Schlegelella brevitalea*, that are notable for the presence of a putrescine appended to the C-terminus. This work appears to build upon the initial detection of glidonins in previous work (Proc. Natl. Acad. Sci. U S A., 115, E4255), for which these molecules have now been structurally elucidated and linked to a near-colinear NRPS.

A huge quantity of work is packed into this manuscript, which includes:

- 1). structural elucidation of glidonins
- 2). Biological testing of glidonins
- 3). identification of enzymes involved in deacetylation and formylation of glidonins – both of which are post-NRPS tailoring steps
- 4). substrate tolerance of the initiation module (A domain (MetOx) and C domain (fatty acyl chain)).
- 5). in vivo and in vitro characterisation of putrescine incorporation steps.
- 6). biological activity testing of glidonin +/- putrescine residue
- 7). domain swapping experiments to incorporate putrescine residues into other lipopeptides
- 8). substrate tolerance of C13 domain for other diamines
- 9). role of acylated form of glidonins in biofilm formation.

As a result, a general comment is that the manuscript is incredibly difficult to digest – with all of the above points discussed in heavy detail. I therefore suggest that sections 2, 3, 6 and 9 are reduced to keep the focus on the ‘biosynthesis’ and ‘engineering’ aspects of the paper, which are the take-home messages.

Response: Thanks for your constructive advice. We deleted and compressed some extra discussions in the above four sections 2, 3, 6 and 9, as well as removed some results to the supplementary notes including “Supplementary Note2: The substrate promiscuity of acylases GdnD”, “Supplementary Note3: Formation of formyl products mediated by a methionyl-tRNA formyltransferase” and “Supplementary Note4: Screening the bioactivity of glidonins”. Two parts “Removal of the fatty acid chains by a PvdQ-like acylase” and “Substrate diversity of the initiation module 1” were integrated into one part “The formation of N-terminal diversity of glidonins”, and the related bioactivity of glidonins was put together in the last part “The bioactivity and function of glidonins”. After revising this manuscript, we focused on the biosynthesis of glidonins and engineering of NRPs to highlight our research point.

See below for specific points on the manuscript:

1). Ln99 – ‘The products of BGC11 in DSM 7029...’ – this is a very inaccessible way to start the results section. Consider revising this opening paragraph for readers who haven’t read (Proc. Natl. Acad. Sci. U S A., 115, E4255).

Response: We revised the section (Characterization of glidonin biosynthetic gene cluster.) and added the description of our previous study, that was “Our previous study showed that the silent NRPS BGC11 from *S. brevitalea* DSM 7029 was activated successfully by *in-situ* constructive promoter P_{Apra} insertion using Redαβ7029 recombineering⁴⁰ and produced a series of products, designated glidonins here, in the activated mutant *S. brevitalea* DSM 7029:P_{Apra}-BGC11 as shown by comparative metabolic profile analysis⁴⁰”.

2). Ln100 ‘Papra’ – should be P_{Apra}

Response: Corrected.

3). Ln113 - which are predicted to activate Phe, Asn, and Pro by A domain specificity

analysis.’ – this needs references.

Response: Done. We added the corresponding reference “(45) Stachelhaus, T., Mootz, H. D. & Marahiel, M. A. The specificity-conferring code of adenylation domains in nonribosomal peptide synthetases. *Chem. Biol.*, **6**, 493-505 (1999).”.

4). HRESIMS – should be ‘HR-ESI-MS’; throughout.

Response: Corrected. We changed HRESIMS to HR-ESI-MS in whole manuscript.

5). Ln157 & Sup Table 10 – please add chromatograms of Marfey’s analysis to the SI to show retention time alignment with authentic standards.

Response: The comprehensive chromatograms of Marfey’s analysis of glidonins was added and shown in the Supplementary Fig. 23.

Supplementary Fig. 23. Marfey’s analysis of the amino acid constituents of glidonins.

6). Ln179 – ‘The final products of Cs domain-embedded’ – please define Cs domains.

Response: We defined a starter condensation domain as a Cs domain, which was mentioned in the introduction section.

7). Ln246 – the ATP/PPi exchange assay is good for assaying initial activation of the amino acid, but not the subsequent loading onto the T domain. Has this step been checked for MetOx and Met for the CsA₁T₁ protein? Perhaps using protein-MS? Also, was the CsA₁T₁ protein obtained from BAP1 cells completely in a holo- form for these assays?

Response: Thanks for your constructive advice. In order to check for Met(O) and Met loading onto the T domain of CsA₁T₁ protein, at first, we used protein-MS as your suggestion. The L-Met, L-Leu and L-Met(O) were used as substrates to load onto the A₁T₁ domain, and HR-ESI-MS analysis was used by followed the method “Supplementary Protocol 3” and “Supplementary Protocol 5a” as mentioned in “(67) Donnelly, D.P., Rawlins, C.M., DeHart, C.J. et al. Best practices and benchmarks for intact protein analysis for top-down mass spectrometry. Nat Methods. 16, 587–594 (2019).” This assay proved that Met, Leu and Met(O) was successfully loaded onto the T₁ domain as we shown in Supplementary Fig. 24.

Meanwhile, to quantify the loading efficiency and also check status of T₁ domain which was purified from BAP1, we conducted hydrolysis and derivatization for the products that was loaded on T₁ domain as we mentioned in the method “***In vitro* assays for the different enzymes of C-terminal multi-domains were performed.**”. In brief, the CsA₁T₁ protein was precipitated and washed the pellet twice by MeOH after loaded the preferred substrates onto CsA₁T₁ domain (purified from BAP1). For testing the loading efficiency, Leu, Met and Met(O) substrates were used with *B. subtilis* phosphopantetheinyl transferase (Sfp) protein separately, and for T₁ domain status test, the Met substrate was loaded with or without Sfp protein. After removed the supernatant and washed the CsA₁T₁ protein pellets, only the loaded substrate was retained. The loaded substrate then was hydrolyzed by KOH and derived by D-FDAA for HR-ESI-

MS analysis. This result also verified that Met, Leu and Met(O) could be loaded onto the T₁ domain (Supplementary Fig. 24e). The loading efficiency of Met was higher once the reaction mixture was supplied with Sfp, which proved that CsA₁T₁ protein obtained from *E. coli* BAP1 cells was not completely in a holo-form as shown in Supplementary Fig. 24f. We thus also supplied Sfp protein for related *in vitro* assay even though the protein was purified from *E. coli* BAP1.

Supplementary Fig. 24. HPLC-MS analysis of proteins A₁T₁ or CsA₁T₁ reacting with substrates. **a.** Structures of A₁T₁ connected to substrates L-Met, L-Met(O) and L-Leu. **b.** HPLC-MS analysis of protein A₁T₁ reacting with substrates. **c, d.** MS/MS profiles of protein A₁T₁ reacting with substrates. **e.** The yield comparison of three substrates linked T₁ domain of CsA₁T₁ protein purified *E. coli* BAP1 through comparing the yield of hydrolyzed target products derivated with FDAA (L-Met-FDAA, L-Leu-FDAA, L-Met(O)-FDAA). The T₁ linked L-Met was quantified as a reference (100%). **f.** The yield comparison of the substrate L-Met linked T₁ domain of CsA₁T₁ protein purified *E. coli* BAP1 through comparing the yield of L-Met-FDAA which was supplied with or without Sfp protein.

8). The ability of the A1 domain to apparently selectively activate Met, MetOx and Leu is quite a strange set of substrate preferences. Have other amino acids been screened against the A1 domain? Some interesting amino acids to test would be: Ile, Val, Ala, Lys based on similarity to residues incorporated into the final product.

Response: We screened the other proteinogenic amino acids, which was shown in Supplementary Fig. 6a. The A₁ domain exhibited broad selectivity to the other amino acids, such as L-Ala, L-Gln, and L-Glu, weaker selectivity for the L-Ile and L-Val, but had no activity for L-Lys. However, the preferred substrate was L-Met.

Supplementary Fig. 6. Substrate selectivity of A₁ domain and Cs domain. **a.** The relative activity of A₁ domain of module 1 for the activation of various amino acids. The activity of CsA₁T₁ for activation of L-Met was quantified as a reference (100%). Data are presented as mean values ±SD (n=3). **b, c.** *In vitro* assay of CsA₁T₁ domains specificity for donor substrates and acceptor substrates at EICs.

9). Ln269/270 – TEMut1, TEMut2 and TEMut1/2 should be written as the amino acid mutations as in Fig 3b. Also, very confused as to which residues have been mutated – presumably one of the Ser residues is the active site (acylated) Ser of a canonical TE domain. What is the role of the other Ser residue? Does the TE domain not have an active site His? This whole section needs to be much clearer.

Response: We changed TE_{Mut1}, TE_{Mut2} and TE_{Mut1/2} to ΔTE₁-S10391A, ΔTE₂-S10624A and ΔTE_{1/2}-S10391A/S10624A as in Fig. 3b, respectively. This whole section was revised carefully and marked in the main text.

Sequence alignment and structural prediction (Supplementary Fig. 20) demonstrate that GdnB-TE domain includes two TE domains (the conserved motif GX~~S~~XG). Through *in vivo* and *in vitro* assays, the results showed that TE₁ and TE₂ domain had hydrolysis function, but did not involve in the release of final product from NRPS assembly line. Because C₁₃ domain could select Put as an intermolecular nucleophile for elongated peptide chain release to generate final product. **The two Ser residues both were active sites of TE domains and essential for their hydrolysis function.** The detailed results and conclusions were shown as follows.

The point mutation of the Ser residue (GdnB-S10391) from TE₁ domain (conserved motif GCSAG) to Ala increased the yield of target product **24** (L-Ala-Put), which indicated this mutation improved the catalytic activity of multidomain protein A₁₂T₁₂C₁₃A*T₁₃TE through *in vitro* assay (Fig. 4b). That means the Ser residue (GdnB-S10391) reduced the hydrolysis function of TE domain. However, this mutation did not influence obviously the production of glidonins *in vivo* (Figs. 3c, 3d). These results the Ser (GdnB-S10391) from TE₁ domain might influence structural stability of the partial protein of GdnB (A₁₂T₁₂C₁₃A*T₁₃TE) *in vitro*, but not whole GdnB protein *in vivo*.

The point mutation of the Ser residue (GdnB-S10624) from TE₂ domain to Ala also increased the yield of target product **24** (L-Ala-Put), which was higher than the Ser residue (GdnB-S10391). This mutation improved the catalytic activity of multidomain protein A₁₂T₁₂C₁₃A*T₁₃TE, which was better than the GdnB-S10391, through *in vitro* assay (Fig. 4b). That means the Ser residue from TE₂ domain had an obvious effect on hydrolysis function of TE than the Ser residue from TE₁ domain. In addition, we could not obtain the multidomain protein A₁₂T₁₂C₁₃A*T₁₃TE_{Mut1/2} containing double mutated sites. Thus, the Ser (GdnB-S10624) from TE₂ domain might not only influence structural stability of the partial protein of GdnB (A₁₂T₁₂C₁₃A*T₁₃TE) *in vitro*, but also the whole GdnB protein *in vivo*, as TE domain from our reported glidomide.

We thus conclude that “TE domain (both for TE₁ and TE₂) retains its hydrolysis function *in vitro* (Fig. 4b), which was further supported by the TE deletion test *in vivo* (Supplementary Fig. 32). The final products can still be obtained for the deletion of TE₂, and the product without Put can be produced for P₁₃-TE (deletion of C₁₃A*T₁₃ and

insertion of the P₁₃ promotor for TE). Despite having a hydrolysis function, the TE domain does not directly participate in the release of final products. Instead, it can impact the folding or stability of the NRPS, as shown both *in vivo* and *in vitro* (Figs. 3d, 4a). This is supported by the observation that deletion of the TE domain and TE₂ significantly decreased the yield of the final products compared with the wild type (Supplementary Fig. 32). Moreover, the TE domain may also assist in the efficient operation of the full-length NRPS pipeline *in vivo* by hydrolyzing any stagnant by-products resulting from the malfunction of the Put release step, as we previously discussed regarding glidomide⁴¹. This was also supported by the observation of an intermediate (**9a**) without the Put moiety found in the crude extract of both wild-type and mutant strain with a constitutive promoter P₁₃⁵³ inserted in front of the TE domain (Supplementary Fig. 32).” as we mentioned in line 279.

The classical TE domain features a conserved catalytic triad Ser-His-Asp. The Ser residue acts as a catalytic nucleophile activated by the His and Asp dyad, which generates an acyl-O-TE covalent enzyme intermediate. GdnB-TE₁ and GdnB-TE₂ domains had no the conserved site His.

Supplementary Fig. 20. Bioinformatic analysis of GdnB-TE domain and structural prediction. **a.** BlastP analysis of the GdnB-TE domain. The red arrow indicates two TE domains. **b.** Sequence alignment of GdnB-TE₁, GdnB-TE₂ and Srf_{TE I} (surfactin). The red frame indicates conserved site residues GX SXG. The red asterisk indicates three catalytic residues Ser, Asp, and His, helices in red, β -strands in blue. **c.** Overall structure of GdnB-TE domain predicted by AlphaFold2, which contains two TE domains. **d.** Structure alignment of TE₁ (green), TE₂ (azure) and the Srf_{TE I} (PDB number: 1JMK) labeled in violet

10). Figure 1 – a few points:

- add a physical space between the GdnA and GdnB subunits.

- add Ppant thiols to all T domains (maybe even add the loaded amino acid structures?)
- add lines over the top of module boundaries
- label putrescine

Response: Thanks for your advice. **Fig.1** was revised in this manuscript according to your suggestions.

Fig 1. The biosynthetic pathway of glidonins. (a) Genetic organization of the *gdn* gene cluster. (b) Organization of modular enzymes involved in the biosynthesis of glidonin. (c) Structures and formation of glidonins A-L (1-12). FMT: methionyl-tRNA formyltransferase, Cs: starter condensation, Cd: dual condensation (C/E), C: condensation, A: adenylation, T: thiolation, TE: thioesterase, A*: truncated adenylation. Cd located on five modules 2, 3, 6, 9, and 11.

Reviewer #2 (Remarks to the Author):

The paper by Chen et al describes the identification of a new NRPS-derived peptide, glidonins, from *Schlegelella brevitalea* DSM 7029.

Unusual features in this biosynthesis are the C-terminal attachment of a putrescin that was described in detail.

In general, I really like the paper, which presents a wealth of information, but I think the paper needs more clarifications at various positions to find the readers way through its content. The starter fatty acid diversity is nothing new and does not need to be presented that prominently. Putrescin as C-terminus was also described before and also the use of terminal amine incorporating domains was shown in detail before in other examples (see below). However, the authors nicely figured out what part of their NRPS was responsible for it.

Experimental details are missing at several parts, as well as structure information for some compounds that were made as controls. Please see my comments below:

Response: We appreciate the reviewer's positive comments and constructive advice. We retained the section of starter fatty acid diversity to clearly clarify the biosynthesis of glidonins in this manuscript. But, this section was not described in detail and the corresponding Figure was added in the supplementary information. We integrated "Removal of the fatty acid chains by a PvdQ-like acylase" and "Substrate diversity of the initiation module 1" into one part "The formation of N-terminal diversity of glidonins".

We added detailed descriptions of related methods marked in blue color, and complete structural information for some compounds in Fig. 4, Supplementary Table 14, Fig. 22, Figs. 26-30, Figs. 32-33.

Supplementary Table 14. The constituents of the synthesized peptides.

Compounds	[M + H] ⁺	Constitutes
1a	1289.6347	D-Met ¹ -D-Asn ² -L-Pro ³ -L-Val ⁴ -D-Phe ⁵ -L-Pro ⁶ -L-Trp ⁷ -D-Val ⁸ -L-Ala ⁹ - D-Ala ¹⁰ -L-Ser ¹¹ -L-Ala ¹²
13a	607.2908	D-Met ¹ -D-Asn ² -L-Pro ³ -L-Val ⁴ -D-Phe ⁵
13	789.4579	lauroyl-D-Met ¹ -D-Asn ² -L-Pro ³ -L-Val ⁴ -D-Phe ⁵
14	789.4579	lauroyl-L-Met ¹ -D-Asn ² -L-Pro ³ -L-Val ⁴ -D-Phe ⁵
15	446.2683	lauroyl-D-Met-D-Asn

10a	1499.8331	myristoyl-D-Met ¹ -D-Asn ² -L-Pro ³ -L-Val ⁴ -D-Phe ⁵ -L-Pro ⁶ -L-Trp ⁷ -D-Val ⁸ -L-Ala ⁹ -D-Ala ¹⁰ -L-Ser ¹¹ -L-Ala ¹²
29a	834.4971	octanoyl-L-Leu-L-Thr-L-Tyr-D-Ala-L-Ala-D-Ala-L-Val
31a	704.4229	octanoyl-L-Val-L-Phe-L-Glu-L-Ile-Gly

Please see my comments below:

- l. 63 ff: there are many more natural products containing C-terminal amines (phenylethylamine, spermidine, tryptamine etc) including putrescine: Cai X et al, Nat Chem 2017; Zhao L et al Org Lett 2018; Paenilamicin Angew Chem 2014, etc

Response: Thanks for your constructive advice. We added the other representative examples of terminal amines, and cited the corresponding references. We revised this sentence to “The additional C-terminal moieties include the amino acid residues that might be mediated by the TE domain, such as bacillothiazols¹⁷ and a threonine-tagged lipopeptide family¹⁸⁻²², and diverse terminal amines, such as diamine putrescine (Put), spermidine (Spe), agmatine (Agm), phenylethylamine (Pea), tryptamine (Tra), and tyramine (Tya)^{14, 23-30}.”

- L 116: can the original specificity of the truncated A domain be suggested/postulated?

Response: The truncated A domain was redefined as the partial A domain (A*). BlastP analysis (Supplementary Fig. 19) showed the partial A domain (A*) only retained the C-terminal catalytic subdomain (A_{sub}), which is responsible for binding AMP. The reference reveals “The A_{sub} domain functions as a flexible hinge whose rotation entails a pull-and-push motion of the adjacent T domain towards and away from the A domain.” Thus, we guessed that the A* domain correlated with downstream T domain might retain GdnB protein structure and had no specificity for substrate, which also confirmed by protein expression.

- l. 117: what is the distance between the two possible active sites?

Response: The distance between the two possible active sites of TE domains (GX SXG) was 227 amino acids. BlastP analysis, sequence alignment and structural prediction showed the GdnB-TE domain contained two TE domains (Supplementary Fig. 20).

- l. 168: lauroylation and lauroyl

Response: We changed lauricyl to lauroyl in whole manuscript.

- l. 203: How was the docking performed in all docking examples in the paper? Please add more details about modelling and docking to the methods or the SI.

Response: Thanks for your suggestion. We added the detailed descriptions into the method “**Rigid receptor docking**” marked in blue color. That was “The structure of GdnD was predicted by AlphaFold2⁵⁴. The structure of the PvdQ in complex with the pyoverdine precursor PVDIq (PDB number: 5UBK) was retrieved from RCSB Protein Data Bank (PDB). GdnD was docked with substrate C14-D-Met(O) in the PVDIq binding pocket of PvdQ using AutoDock Vina. To cover the entire binding site, a 2 nm side cubic grid box was positioned at the geometric center of residues S486 (PvdQ S268), N497 (PvdQ D269), R516 (PvdQ R297), and H683 (PvdQ H457) as reported⁷¹. A total of 20 poses for one substrate were generated through the docking procedure, and the complex that closely matched the binding pose of PVDIq in PvdQ was selected independently for further analysis.

The docking procedure for the C₁₃ domain with Put followed a similar approach. RzmA mutant H140A/R148A in complex with C8-CoA and Leu-SNAC (PDB number: 7C1S) was used as the model complex and 2 nm side cubic grid box was positioned at the geometric center of residues H9721 (RzmA H140), T9879 (RzmA T289), V9877 (RzmA S287) and L9860 (RzmA P271) as reported⁵⁶. The complex that closely matched the binding pose of Leu-SNAC in RzmA and 4'-phosphopantetheine in the condensation domain of holo-AB3403 (PDB number: 4ZXI⁷²) was selected for further analysis.”

- L 219 ff: the finding of a formylated peptide in *E. coli* is interesting but is this relevant in the original producer? Were formulated products also found there? If not, please remove this part from Fig. 1 since it might not be part of the pathway. If the formyl derivatives are found in the DSM strain, the authors might be able to perform an *in vitro*

assay with the methionyl-tRNA formyltransferase as most likely candidate for the modification.

Response: Thanks for your considerable advice. We isolated and identified the formylated products **4** and **5** from the activated mutant DSM 7029:P_{Apra}-*gdn*, and detected a formylated product **4** in *E. coli* BL21. The results ensured the formation of formylation was mediated by a formyltransferase from native DSM 7029 and *E. coli*. The genome of DSM 7029 included three formyltransferases PurN, PurH, and methionyl-tRNA formyltransferase. We deleted successfully PurN and PurH except methionyl-tRNA formyltransferase. Metabolic analysis showed two mutants could still produce glidonins, suggesting the formylation might be catalyzed by methionyl-tRNA formyltransferase. *In vitro* assay showed that purified methionyl-tRNA formyltransferase could catalyze exogenous peptides **1a** and **13a** to generate the formyl products **1b** and **13b**, and enhance the yield of formyl product (**4**) of **1** compared with control (**1** contains a trace of **4** in the control) (Supplementary Fig. 22). These results were presented in Supplementary Note3“**Formation of formyl products mediated by a methionyl-tRNA formyltransferase**”. The related procedures were added in the method section “The *in vitro* assay of formyltransferase was tested⁶⁸.”

Supplementary Note3: Formation of formyl products mediated by a methionyl-tRNA formyltransferase

N-formylation is a rare event in NRPs, even though it is a common mechanism for initiation of ribosomal protein biosynthesis in prokaryotes.⁹ The *N*-terminal formylation of three NRPs gramicidin¹⁰, anabaenopeptilides¹¹ and szentiamide^{12,13} was catalyzed by *N*-terminal formylation (F) domain, respectively. However, the lack of F domain in the *gdn* BGC hints that the formation of the *N*-formylation of glidonin is clearly different with the above three NRPs. Genome analysis showed the genome of *Escherichia coli* BL21 and *Schlegelella brevitalea* DSM 7029 both included three common formyltransferases family containing phosphoribosylglycinamide formyltransferase (PurN), phosphoribosylaminoimidazolecarboxamide formyltransferase (PurH), and methionyl-tRNA formyltransferase, suggesting the

formyl moiety of **4** might be catalyzed by the formyltransferase(s) (Supplementary Table 13). Then, we successfully constructed two knock-out mutants of *purN* and *purH* with the fragile growths, but failed to obtain the inactivation of methionyl-tRNA formyltransferase. The metabolite analysis showed the two mutants still could produce the glidonins (Supplementary Fig. 22e), suggesting the formylation might be catalyzed by methionyl-tRNA formyltransferase (defined as Fmt protein). Subsequently, we purified this protein, and carried out *in vitro* assay of protein Fmt with different substrates **1**, **1a**, and **13a**. LC-MS profiles showed that the yield of formyl product (**4**) of **1** was higher than control (**1** was mixture containing a trace mass of **4**), and the other formyl products **1b** and **13b** also could be detected in reaction mixture, respectively (Supplementary Figs. 22b, 22c, 22d). Thus, the formation of formyl moiety was catalyzed by methionyl-tRNA formyltransferase.

Supplementary Fig. 22. The structures of lipopeptides and NRPs, and *in vitro* assay of Methionyl-tRNA formyltransferase (Fmt). **a.** The structures of synthesized lipopeptides, and corresponding nonacylated compounds and formylated compounds. **b.-d.** HPLC-MS analysis of reaction mixture of protein Fmt with different substrates **1**, **1a** and **13a**. **4** was formyl product of **1**, **1b** was formyl product of **1a**, **13b** was formyl product of **13a**. **e.** LC-MS analysis of crude extracts of the activated mutant DSM 7029:P_{Genta-gdn} and two deleted mutants of formyltransferases PurH and PurN.

- L 228: Formylation in szentiamide: see Bode et al Angew Chem 2019 (SI)?

Response: Thanks. We revised this sentence “The N-terminal formylation of three

NRPs gramicidin¹⁰, anabaenopeptilides¹¹ and szentiamide^{12,13} was catalyzed by N-terminal formylation (F) domain in the corresponding NRPSs, respectively.” and added the corresponding reference “(13) Bode, E., Heinrich, A. K., Hirschmann, M., et al. Promoter activation in Δhfq mutants as an efficient tool for specialized metabolite production enabling direct bioactivity testing. *Angew. Chem. Int. Ed. Engl.*, **58**, 18957-18963 (2019).” We moved this part to the Supplementary Note3 “Formation of formyl products mediated by a methionyl-tRNA formyltransferase” as the reviewer1’s suggestion.

- L 289: Why was **24** derivatized with D-FDAA and what is the structure of **24a**? Where is this described in the methods?

Response: The structure of **24a** composes of L-Ala-Put-FDAA shown in Fig. 4b, which was from target product **24** (L-Ala-Put) derivatized with D-FDAA. The target product **24** (L-Ala-Put) was from condensing between substrate L-Ala and Put mediated by C-terminal enzymes of GdnB, which was difficult to detect directly. Thus, we employ **24** derivatized with D-FDAA for detecting easily by LC-MS analysis. The other target products **25**, **26**, **27** also apply this method, and structures of the three derivatives **25a**, **26a** and **27a** were shown in Supplementary Fig. 31a.

The procedure of the derivatized reaction was described in detail in the methods (Marfey’s analysis of the constitutes of glidonins and Purification and characteristics of enzymes). That was “**Marfey’s analysis of the constitutes of glidonins**. Approximate 0.5~1 mg of compounds was hydrolyzed with 6 M HCl (0.5 mL) at 90 °C for overnight. The solutions were dried by nitrogen, and the mixture was dissolved in 200 μ L of H₂O. The standard amino acids were also dissolved in 200 μ L of H₂O. Then, the above samples were added 1 M NaHCO₃ (25 μ L) and 1% 1-fluoro-2,4-dinitrophenyl-5-L-alanine amide (L-FDAA, 200 μ L)⁴⁹, and the mixture was vortexed and incubated at 40 °C for 60 min. Then, 100 μ L of 2 M HCl was added to quench the reaction and the filtered solution was analyzed by LC-MS using AcclaimTM RSLC 120 C₁₈ column (2.1 \times 100 mm, 2.2 μ m) with a linear gradient of ACN with elution conditions (5%-50% ACN in 35 min) at a flow rate of 0.3 mL/min and UV detection at 330 nm. The LC-MS condition:

0-3 min, 5% ACN; 3-38 min, 5%-50% ACN; 38.1 min, 95% ACN; 38.1-42 min, 95% ACN; 42.1 min, 5% ACN; and 42.1-45 min, 5% ACN, ACN and H₂O contained 0.1% FA.” as mentioned in line 658, and “The reaction mixture was dissolved with 200 μ L ddH₂O, and then the mixture was derivated with D-FDAA according to the procedure of the above marfey’s analysis. The final reaction mixture was suspended using 100 μ L 2 M HCl, which subsequently was dried by SpeedVac. Finally, the mixture dissolved with 100 μ L MeOH to detect the target products (**24a**, **25a**, **26a**, and **27a**) using LC-MS analysis (5% 95% ACN in 15 min).” as mentioned in line 732.

- L 345: do the authors know if the compounds with and without putrescine address the same target or can they measure uptake into the cell? If the state about the improvement of the bioactivity they imply the same target of both compounds but it can also be different targets.

Response: Thank you for this comment. We carried out the relative experiments, which shown in Supplementary Note4 “Screening the bioactivity of glidonins and derivatives”. The description was “To determine whether the difference in bioactivity of products (with or without Put moiety) is due to variations in cellular uptake, we treated hepatoma cell line HepG2 with compounds **7** (C12 acyl), **10** (C14 acyl), and **10a** (C14 acyl and without Put) (Supplementary Fig. 33c). The cells were treated for 3 h, and then culture medium and cells were extracted separately. LC-MS was used to detect and quantify the compounds. At lower concentrations (5000 ng/mL, and 10,000 ng in total for each group), and shorter periods of treatment, none of the three products showed inhibition of cells (Supplementary Fig. 33a). The results suggest that the compounds with strong activity (**7** and **10**) have lower total quantities (intracellular plus extracellular) compared to **10a**, indicating that differences in cell metabolism between

the two compounds (**10** vs **10a**) may be contributing to the differences in antitumor activity (Supplementary Fig. 33b). This could be due to differences in the speed of cellular uptake or differences in drug metabolism rate, requiring further experimental investigation.” More detailed experiments in the future will be performed to elucidate the reason.

Supplementary Fig. 33. Analysis of hepatoma cell line HepG2 treated with compounds **7** (C12 acyl), **10** (C14 acyl), and **10a** (C14 acyl and without Put). **a.** Analysis of cell numbers treated with compounds. **b.** LC-MS analysis of extracellular and intracellular compounds. **c.** Structures of compounds **7**, **10**, and **10a**.

- L 357/374: what is the structure of **29** and **29a** and **31** and **31a**? How were they synthesized and confirmed? Why was NO production measured? Explain in more detail also in the methods.

Response: The lipopeptides **29a** and **31a** were synthesized by Sangon Biotech company, and the correct structures of them were confirmed by HR-ESI-MS, ^1H and ^{13}C spectrum shown in Supplementary Fig.26, Figs.103-106.

We measured NO production to test the anti-inflammatory activity of compounds and their derivatives. The detailed procedure of measurement of NO production was described in the method section “Bioactivity assays screening of glidonins and derivatives” in line 764, which was “RAW 264.7 cells were seeded in 96-well plates and incubated for 24 h. Then, cells were treated with compounds **29**, **29a**, **31** and **31a** (0-80 μ M) for 30 min, followed by co-treatment with LPS (10 mg/mL) for another 24 h. The Nitric Oxide assay kit with Griess reagents (Beyotime, Lot: S0021, Shanghai, China) was used to examine cellular supernatant nitrite accumulation, which represents cellular NO levels. The OD value of 540 nm absorbance was detected with a microplate reader.”

Supplementary Fig. 29. The structures and anti-inflammatory bioactivity of lipopeptides. **a.** Structures of lipopeptides **29a**, **29**, **31a**, and **31**. **b.** The effects of compounds on NO production in LPS-induced RAW264.7 cells between **29**, **31** and each Put-lacking derivatives **29a**, **31a**. NO production in the absence of LPS was used as negative control. LPS-stimulated NO production in the absence of compounds was used as the positive control.

- L 389 ff: the use of terminal amine incorporating domains to generate different amine derivatives was already shown extensively in Cai et al, Nat Chem 2017. What is the bioactivity of the non-natural derivatives?

Response: Cai et al. demonstrated that the terminal amine influence lipopeptide activity against protozoa and insects according to rhabdopeptides and xenortides were active

against different protozoa and insects. Based on our results that the engineering derivatives containing Put moiety exhibit moderate anti-inflammatory activity (NO production) compared with lipopeptides without Put moiety. It is reasonable to infer that diamine derivatives with the same backbone should have similar activity. The purpose of substrate additions in this study is to showcase the potential of engineering the “Put unit”, the activity of the unnatural product was not measured here due to the low yields.

- L 442 f: How does the biofilm modulation fits to the also described cytotoxic activity?

Response: This is a thoughtful question. The biofilm protects bacteria from unsuitable environment. But, the basic mechanisms of biofilm regulation are not clear enough to date, even though structural similarity of lipopeptides does not indicate if they are involved in formation or degradation of biofilm. The acylated glidonins improve strain’s biofilm formation and have cytotoxic activity against tumor cells, but deacylated glidonins do not. The biofilm formation and cytotoxic activities might be correlated with the fatty acid chains of glidonins. That means the biofilm modulation may be fits to the cytotoxic activities although their targets are quite different, which might depend on the fatty acid chains attached to cell members.

- Fig. 1: Explain in the text more about the truncated A domain or show an alignment in the SI so one can see what happened here?

Response: The blastP analysis of module 13 including the truncated A domain (redefined as partial A domain to avoid misunderstand) was shown in Supplementary Fig. 19. Moreover, the sequence alignment showed that the partial A domain was an AMP-binding C, which was a C terminal catalytic subdomain A_{sub}.

Supplementary Fig. 19. Bioinformatic analysis of Gdn NRPS gene cluster. **a.** AntiSMASH prediction of Gdn NRPS gene cluster. **b.** BlastP analysis of the C₁₃A*T₁₃ domains of the GdnB. **c.** Sequence alignment of GdnB-A* and 1AMU_A. The red frame indicates Stachelhaus code.

- Fig. 2. In d-g: show both compounds, substrate and product in the chromatograms as in c. Change lauricyl to lauroyl. What is the substrate specificity regarding the amino acid composition? The authors have tested Met only but can include other amino acids as well. Where are the structures of **16** and **17**? All numbers must be reflected by structures either in the text or the SI.

Response: Thanks for your suggestions. We revised Fig. 2 and its legend, which includes substrates and products in the chromatograms, and changed the lauricyl to lauroyl in the legend of Fig. 2.

The substrate specificity of GdnD was widespread whatever the configurations and lengths of peptides according our results. The PvdQ could deacylated the fatty acid chain of the compounds **7** (D-Met), **8** (D-MetO), and **17** (L-Leu), which was shown in Fig. 2c and Fig. 2d, that indicated the other amino acids as substrates could be deacylated combined with the “V”-shaped structure of GdnD. And we added the

detailed description about the specificity of GdnD in Supplementary Note2 “The substrate promiscuity of acylase GdnD” as reviewer3’s suggestion.

The relative structures of compounds (**13**, **13a**, **13b**, **14**, **14a**, **14b**, **15**, **15a**, **16**, **17**, and **17a**) were added in the supplementary Table and Figures (Supplementary Table 14, Fig. 22).

Fig 2. Identification of Cs domain and acylase PvdQ enzymes. (a) LC-MS analysis of crude extracts at EIC 650-800 of activated mutant DSM 7029P_{Apra-gdn} and two inactivated mutants of Cs domain, DSM 7029P_{Apra-gdn:CsHH::AA} and DSM 7029P_{Apra-gdnΔCs}. (b) HPLC-MS profile of crude extracts of activation mutant DSM 7029:P_{Genta-gdn}, and three deletion mutants of acylases PvdQ. PvdQ-00: AKJ26698.1, GdnD: AKJ30942.1, PvdQ-02: AKJ29580.1. (c)-(g) Biochemical characterization of the acylase GdnD-catalyzed reactions by LC-MS analysis. **13**: lauroyl-D-Met-D-Asn-L-Pro-L-Val-D-Phe (EIC 789 [$M + H$]⁺), **13a**: D-Met-D-Asn-L-Pro-L-Val-D-Phe (EIC 607 [$M + H$]⁺), **14**: lauroyl-L-Met-D-

Asn-L-Pro-L-Val-D-Phe (EIC 789 [$M + H$]⁺), **14a**: L-Met-D-Asn-L-Pro-L-Val-D-Phe (EIC 607 [$M + H$]⁺), **15**: lauroyl-D-Met-D-Asn (EIC 446 [$M + H$]⁺), **15a**: D-Met-D-Asn (EIC 264 [$M + H$]⁺), 16: Rzm A (Rhizomide A) (EIC 732 [$M + H$]⁺), 17: C14-Rzm A (C14-Rhizomide A) (EIC 900 [$M + H$]⁺), **17a**: deacylation of C14-Rzm A (EIC 690 [$M + H$]⁺), **4**: glidonin D (EIC 693.2 [$M + 2H$]²⁺), **13b**: formyl-D-Met-D-Asn-L-Pro-L-Val-D-Phe (EIC 633 [$M + H$]⁺), **14b**: formyl-L-Met-D-Asn-L-Pro-L-Val-D-Phe (EIC 633 [$M + H$]⁺).

- Fig. 4: explain the docking method

Response: The docking method was shown in the method section “**Rigid receptor docking**”.

Rigid receptor docking. The structure of GdnD was predicted by AlphaFold2⁵⁴. The structure of the PvdQ in complex with the pyoverdine precursor PVDIq (PDB number: 5UBK) was retrieved from RCSB Protein Data Bank (PDB). GdnD was docked with substrate C14-D-Met(O) in the PVDIq binding pocket of PvdQ using AutoDock Vina. To cover the entire binding site, a 2 nm side cubic grid box was positioned at the geometric center of residues S486 (PvdQ S268), N497 (PvdQ D269), R516 (PvdQ R297), and H683 (PvdQ H457) as reported⁷¹. A total of 20 poses for one substrate were generated through the docking procedure, and the complex that closely matched the binding pose of PVDIq in PvdQ was selected independently for further analysis.

The docking procedure for the C₁₃ domain with Put followed a similar approach. RzmA mutant H140A/R148A in complex with C8-CoA and Leu-SNAC (PDB number: 7C1S) was used as the model complex and 2 nm side cubic grid box was positioned at the geometric center of residues H9721 (RzmA H140), T9879 (RzmA T289), V9877 (RzmA S287) and L9860 (RzmA P271) as reported⁵⁶. The complex that closely matched the binding pose of Leu-SNAC in RzmA and 4'-phosphopantetheine in the condensation domain of holo-AB3403 (PDB number: 4ZXI⁷²) was selected for further analysis.”

- Fig. 6. Explain the use of D-FDAA in methods. Isolate and test the derivatives or move whole figure into the SI since the results are not too surprising but expected from previous work by others (see above).

Response: Thanks for your suggestion. The procedure of the target products derivatized

with D-FDAA for detecting easily by LC-MS analysis was described in detail in the method (Purification and characteristics of enzymes). We moved the Fig. 6 into the Supplementary Fig. 31.

“The reaction mixture was dissolved with 200 μ L ddH₂O, and then the mixture was derived with D-FDAA according to the procedure of the above marfey’s analysis. The final reaction mixture was suspended using 100 μ L 2 M HCl, which subsequently was dried by SpeedVac. Finally, the mixture dissolved with 100 μ L MeOH to detect the target products (**24a**, **25a**, **26a**, and **27a**) using LC-MS analysis (5% 95% ACN in 15 min).” as mentioned in line 732.

SI:

- Give solvent and field for all NMR data in Tables and figures

Response: Done. The solvents MeOH-*d*₄ and DMSO-*d*₆ for compounds were added in Supplementary Tables and Figures.

- Fig S1. Give wt control as well

Response: Done. We added WT control in the Fig. S1 and revised the Fig. S1 and its legend.

Supplementary Fig. 1. The HPLC-MS profile of the products of glidonin gene cluster in the activated mutant DSM 7029: P_{Apra} -*gdn* and DSM 7029 wild type.

- S2c: the chromatogram looks rather different than that in S1. Please explain and clarify

in methods.

Response: We used UHPLC-HR-MS and LC-MS for detecting the crude extracts, and the relative conditions of two methods were shown in method “Metabolic analysis.” That was “The UHPLC system was performed using an ODS column (Luna RP-C₁₈, 4.6×250 mm, 5 μm, 0.75 mL/min) with gradient elution. Mass spectra was acquired in centroid mode ranging from 100 to 1500 *m/z* with positive-mode electrospray ionization and auto MS² fragmentation. HPLC parameters were as follows: solvent A, H₂O with 0.1% formic acid (FA); solvent B, 0.1% FA in acetonitrile (ACN); gradient at a constant flow rate of 0.75 mL/min, 0-5 min, 5% B; 5-55 min, 5%-95 % B; 55.1 min, 95% B; 55.1-60 min, 95% B; detection by UV spectroscopy at 200-400 nm. The HPLC-MS or LC-MS system was performed using Acclaim™ RSLC 120 C₁₈ column (2.1×100 mm, 2.2 μm) and its condition was: 0-3 min, 5% ACN; 3-18 min, 5%-95% ACN; 18.1 min, 95% ACN; 18.1-22 min, 95% ACN; 22.1 min, 5% ACN; and 22.1-25 min, 5% ACN, ACN and H₂O contained 0.1% FA; flow rate by 0.3 mL/min; detection by UV spectroscopy at 200-400 nm.”

- S3: Show all peptides linear from N-C from left to right (in all Figures also in S9 etc). Do not bend the molecules since it makes it more difficult to read. 9 is missing part of the aromatic ring.

Response: Thanks for your suggestion. We revised the structures of Supplementary Fig. 3 and the structures of other compounds in whole manuscript.

Supplementary Fig. 3. Key COSY and HMBC correlations of **1**, **2**, **4**, **6-10**, **29**, and **31** (Black: Signals were recorded in MeOD-*d*₄, Red: Signals were recorded in DMSO-*d*₆).

- S8: ad domain organization, otherwise one can not understand the text on line 263.

Response: We added the domain organization on the different gene clusters including RXP gene cluster in the Supplementary Fig. 8, and revised its legend.

Supplementary Fig. 8. Gene organization of eight nonribosomal peptide synthetase (NRPS) gene clusters. orb: ornibactin, mba: malleobactin, cro: crochelin, cph: cepaciachelin, bic: bicornutin A1, odl: odilorhabdin (NOSO-95), inx: Rhabdopeptide/xenortide-like peptides (RXPs), gdn: glidonin. Red indicates NRPS core genes, blue indicates additional genes, grey indicates non-related genes. C: condensation domain, A: adenylation domain, T: thiolation domain, E: epimerization domain, TE: thioesterase domain, MT: methyltransferase domain, Cal: acyl-CoA ligase domain, KR: ketoreductase domain, KS: ketoacylsynthase domain, AT: acyltransferase domain.

- S16: give more details in the text/methods about the modelling and the performed analysis.

Response: We removed MD analysis based on reviewer 3' comment. We directly

mutated S9663, D9665 and E9984 to other amino acids (S9663A, S9663V, S9663Q, D9665A, D9665L, D9665E, E9984A, E9984M, E9984D) for testing the function of three residues, and the results indicate that the three crucial residues are located within the binding pocket, and are essential for interacting with substrate Put. The detailed method and results were shown in manuscript.

The results were that “The *in vitro* assays showed that the yield of **24a** from three mutants (S9963A, D9965A and E9984A) was greatly decreased, almost to that of A₁₂T₁₂, at the premise of their similar activity toward the substrate L-Ala (Fig. 4d, Supplementary Figs. 16a, 16c). Moreover, the three mutants altered the substrate specificity of the C₁₃ domain for the substrates putrescine and 1,5-diaminopentane, resulting in a reduced catalytic activity toward putrescine and increased catalytic activity toward 1,5-diaminopentane (Supplementary Fig. 16b), indicating that these three crucial residues are located within the binding pocket. The recovery of catalytic activity in the mutants with the original amino acid properties (S9963A vs. S9963Q, D9965A vs. D9965E and E9984A vs. E9984D) suggested the importance of the properties of these three residues (Fig. 4d). This indicated that these three residues might be key sites for substrate binding, especially E9984, which is located on the β -sheet of the C₁₃ domain. The impact of the length of the amino acid side chain on the product yield (S9963A vs. S9963V, D9965A vs. D9965L and E9984A vs. E9984M) suggested that it also created steric hindrance in the substrate binding pocket (Fig. 4d). In summary, the three residues were essential for interacting with substrate Put, and our mutation results indicated that engineering the substrate specificity of this releasing module is a viable strategy.”

- NMR data: I do not find the labelling of the peaks and the integrals very useful in the spectra since there is a table with all this data. It would be nicer to assign these signals or leave the label out and refer to the tables only (but still show the spectra).

Response: Corrected. We revised the spectra of the supplementary Figures according to your suggestion.

Reviewer #3 (Remarks to the Author):

This manuscript reports the discovery of glidonins and their biosynthesis and biological activities. A lot of data were presented. The role of the termination module in putrescine incorporation is perhaps most interesting. A C13 domain was suggested to catalyze the incorporation, and the TE domain was somehow also involved. This termination module was swapped into other nonribosomal peptide synthetases and used to introduce putrescine into the termini of two different peptides. The difference in bioactivity between the acylated and deacylated variants is also interesting. However, the precise role of the TE domain in putrescine incorporation was not determined, and the efficiency of the engineered pathways did not take incomplete or shunt products into consideration. Description of several experiments lacks key information, such as concentration of compound used.

Overall, although new insights were made on a silent gene cluster, the manuscript is poorly written with many grammatical and spelling mistakes, it lacks statistical analysis, and the conclusions are diffused. Specific comments are included below:

Response: We are grateful for the positive comments provided by the reviewer. We conducted additional detailed experiments and analyses on the TE domain to address the main concern raised in the evaluation. As we concluded in line 279: “the TE domain (both for TE₁ and TE₂) retains its hydrolysis function *in vitro* (Fig. 4b), which was further supported by the TE deletion test *in vivo* (Supplementary Fig. 32). The final products can still be obtained for the deletion of TE₂, and the product without Put can be produced for P₁₃-TE (deletion of C₁₃A*T₁₃ and insertion of the P₁₃ promotor for TE). Despite having a hydrolysis function, the TE domain does not directly participate in the release of final products. Instead, it can impact the folding or stability of the NRPS, as shown both *in vivo* and *in vitro* (Figs. 3d, 4a). This is supported by the observation that deletion of the TE domain and TE₂ significantly decreased the yield of the final products compared with the wild type (Supplementary Fig. 32). Moreover, the TE domain may also assist in the efficient operation of the full-length NRPS pipeline *in vivo* by hydrolyzing any stagnant by-products resulting from the malfunction of the Put release step, as we previously discussed regarding glidomide⁴¹. This was also supported by the

observation of an intermediate (**9a**) without the Put moiety found in the crude extract of both wild-type and mutant strain with a constitutive promoter P₁₃⁵³ inserted in front of the TE domain (Supplementary Fig. 32).

We apologize for the errors in the manuscript and have taken steps to improve the writing quality and accuracy of the experimental descriptions, including adding missing information such as compound concentrations. Regarding the consideration of incomplete or shunt products, we will revise the manuscript to better address this point as we included in below question. Thank you again for your valuable feedback.

Supplementary Fig. 32. Analysis of TE domain influencing the glidonin biosynthesis. **a.** HPLC-MS analysis of crude extract of four mutants compared with wild type. The yield of **9** generated from the wild type strain DSM 7029:P_{Apra}-gdn was quantified as a reference (100%). N.D. indicates no products. **b.** The diagram of different mutants. Promoter P₁₃ was from DSM 7029 genome. **c.** Structures of **9** and intermediate **9a**.

1. Line 108-109, “while the other mutants fail to impact on the production of glidonin” is misleading. Looks like mutants of *gdnA* and *gdnB* were not made.

Response: The two core NRPS genes *gdnA* and *gdnB* were essential for biosynthesis of glidonins, and were deleted resulting in the absence of glidonin, which were shown in the part “The formation of N-terminal diversity of glidonins” and “Assembly of the Putrescine moiety”. This part was to clarify if the additional genes are involved in the biosynthesis of glidonins. So, we did not present the LC-MS analysis of deletion of the core genes in here.

2. Line 117, sequence alignment that shows the two conserved active sites in the TE domain should be included in the SI.

Response: The sequence alignment between two TE domains and Srf_TEI were shown in supplementary Fig. 20. Also, the blastP analysis and structural prediction of TE domain showed the TE domain contained two TE domains (Supplementary Fig. 20).

Supplementary Fig. 20. Bioinformatic analysis of GdnB-TE domain and structural prediction. **a.** BlastP analysis of the GdnB-TE domain. The red arrow indicates two TE domains. **b.** Sequence alignment of GdnB-TE₁, GdnB-TE₂ and Srf_TE I (surfactin). The red frame indicates conserved site residues GX₁SX₂G. The red asterisk indicates three catalytic residues Ser, Asp, and His, helices in red, β -strands in blue. **c.** Overall structure of GdnB-TE domain predicted by AlphaFold2, which contains two TE domains. **d.** Structure alignment of TE₁ (green), TE₂ (azure) and the SrfTE I (PDB number: 1JMK) labeled in violet.

3. Line 138, “rotamer” was not the correct term to use here.

Response: Corrected. We changed “rotamer” to “isomer”.

4. Figure 1b, would be helpful to specify which ones are the dual C domains.

Response: We added “Cd domain on the five modules 2, 3, 6, 9 and 11” to the legend of Fig. 1.

5. Line 177, the authors mentioned that glidonins had no antimicrobial activities, but later on in Figure 7c, **7** and **10** were found to inhibit the growth of the producing strain. This seems conflicting, and the names of the bacterial strains tested should be provided.

Response: Thanks for your advice. The antimicrobial activity for seven tested strains employed inhibition zone, which include four tested bacteria (*Escherichia coli*, *Pseudomonas aeruginosa*, *Staphylococcus aureus* and *Bacillus subtilis*), and three tested fungi (*Candida albicans*, *Rhizoctonia solani* and *Aspergillus fumigatus*). These seven strains were presented in the supplementary Note4: “Screening the bioactivity of glidonins”. The acylated compounds (**7** and **10**) exhibit no inhibition to native host DSM 7029 employing the same method as we tested. The method needs increased bioactivity of compounds.

However, compounds **7** and **10** exhibited weak inhibition of growth of wild type DSM 7029, whose OD₆₀₀ was 0.6 (for control) and OD₆₀₀ was 0.4 (for addition of compounds). That did not lead to form the inhibition zone. Thus, we employed two methods for studying the bioactivities of glidonins.

6. Line 270, Fig. 3e was referenced, but there is no panel e in Figure 3.

Response: Corrected. We changed Fig. 3d, 3e to Fig. 3c, 3d.

7. Are the same residues in PvdQ conserved in GdnD for substrate binding? What are the sequence similarity between the two proteins? Rationalization of the promiscuity of GdnD was vague.

Response: The sequence similarity between PvdQ proteins and GdnD was 68% and shown in Supplementary Fig. 5d. The red asterisk indicates the substrate binding residues, and the green asterisk indicates the catalytic residues. We revised this part and moved it to the supplementary information which was “Supplementary Note2: The

substrate promiscuity of acylase GdnD”.

The detailed description was “We investigate the substrate specificity of GdnD. At first, we predicted the structure of GdnD using AlphaFold2¹, which is “V”-shaped structure, and its substrate pocket is open that can accommodate broad substrates (Supplementary Figs. 5b, 21a). The biochemical assays showed the nonacylated products **13a**, **14a** and **15a** were detected from reaction mixing with three glidonin-like shorter lipopeptides **13**, **14** and **15**, respectively, indicating GdnD can hydrolyse lipopeptides with L-/D-configuration of first amino acids and different lengths of peptide chains from dipeptide to dodecapeptide (Fig. 2c, 2e, Supplementary Table 14, Fig. 22a). Moreover, GdnD was functional for other exogenous lipopeptides except short acyl chains because the GdnD could remove myristoyl of cyclic lipopeptide C14-rhizomide A (**17**) to yield nonacylated product **17a** but not acetyl of rhizomide A (**16**) (Fig. 2f, Supplementary Fig. 22a).

The PvdQ shows a heart-shaped structure (“V”-shaped structure) formed from two interleaved chains, and a hydrophobic pocket that presents the catalytic Ser β 1 responsible for the acyl hydrolysis.^{2,3} The residues of lining of substrate-binding site are highly conserved among PvdQ homologues.⁴ Numerous studies have reported that PvdQ acylases can hydrolyze various substrates, including different N-acyl-homoserine lactones (HSLs)⁵, telomycins (C8-C12)⁶, and some siderophores crochelins (C10-C12)⁷ and marinobactins (C12-C16)⁸, whereas the concrete mechanism of PvdQ inclined to broad substrates is not the main focus. But, the substrate promiscuity of acylase GdnD was investigated based on the reported PvdQ structures.

The crystal structures of PvdQ (PDB number 5UBK and 3L91) show that the protein adopts a “V”-shaped structure, with a hydrophobic pocket located at the vertex of the “V” that can accommodate N-terminal different fatty acid chains of substrates (Supplementary Figs. 5b, 5c)^{2,4}. According to these characteristics, the substrates are disfavoured when the N-terminal length is either too short to reach the hydrophobic pocket or too long to fit inside it (Supplementary Fig. 5c)⁵. Residues W400 (W402 in PvdQ), V284 (V286 in PvdQ), F238 (F240 in PvdQ), F246 (F248 in PvdQ), L264

(L266 in PvdQ), W377 (W378 in PvdQ), L267 (L269 in PvdQ), M373 (V374 in PvdQ), V401 (V403 in PvdQ), form a hydrophobic pocket in both GdnD and PvdQ structures (Supplementary Fig. 21c). Thus, we speculated that GdnD might follow a similar principle when it hydrolyzed lipopeptides with varying lengths of peptide chains. In addition, GdnD can hydrolyse lipopeptides with both L-/D- configuration of the first amino acid and various amino acids in the backbone. This observation could be attributed to the accessibility of the entrance of the “V”-shaped pocket due to its open conformation at C-terminus (Supplementary Fig. 21a, 21b). The binding pocket for the substrate’s C-terminus, which is located at the entrance of the “V”-shaped pocket, is relatively open and can accommodate larger and different C-terminal groups to enter. In summary, the binding pocket for the C-terminus shows a wider substrate promiscuity compared to the binding pocket for the N-terminus.”

Supplementary Fig. 21. Analysis of GdnD docking with substrate C14-D-Met(O). **a.** GdnD docking with C14-D-Met(O), rectangles in blue and red represent the C-terminus and N-terminus of substrate. **b.** Zoom view of the binding pocket of GdnD. **c.** The hydrophobic pocket in C-terminus, S215 was the active site of GdnD which located in N-terminus.

8. No statistical significance was shown in Figure 4 or Figure 7 to compare the activities of the different mutants.

Response: Done.

9. The three residues identified in C13 are in a loop region and unlikely to be accurate in the model. It is possible that the alanine mutations at this loop region affected the local folding and lost activity. Conclusion drawn from RMSD analysis is very speculative.

Response: Thank you for your helpful suggestion. We have taken your advice and removed the MD analysis to avoid any confusion. In regards to the three residues (S9963, D9965, and E9984), E9984 was found in the β -sheet of the C₁₃ domain, while the other two were located in the loop region. To confirm that the point mutations were affecting the binding pocket and not just the local folding of the protein, we conducted catalytic activity measurements using an unnatural substrate (1,5-diaminopentane, its product **25**) as a comparison to the original substrate (Put, its product **24**) (Supplementary Fig. 31a). Any changes in substrate specificity would indicate that these three residues are indeed located within the substrate binding pocket. As we mentioned in line 305: “Moreover, the three mutants altered the substrate specificity of the C₁₃ domain for the substrates putrescine and 1,5-diaminopentane, resulting in a reduced catalytic activity toward putrescine and increased catalytic activity toward 1,5-diaminopentane (Supplementary Fig. 16b), indicating that these three crucial residues are located within the binding pocket.”

Supplementary Fig. 16. Function analysis of A₁₂T₁₂C₁₃A*T₁₃ and its mutants. **a.** The relative activity of A₁₂ domain from proteins A₁₂T₁₂C₁₃A*T₁₃, its three mutants and A₁₂T₁₂ for the substrate L-Ala. The activity of A₁₂T₁₂C₁₃A*T₁₃ for activation of L-Ala was quantified as a reference (100%). Data are presented as mean values ±SD (n=3). **b.** Yield comparison of the target products **24a** and **25a** obtained from A₁₂T₁₂C₁₃A*T₁₃ and its three mutants for testing the condensation activity to substrates putrescine and 1,5-diaminopentane. The yield of **24a** and **25a** generated from the wild-type A₁₂T₁₂C₁₃A*T₁₃ was quantified as a reference (100%), respectively. Data are presented as mean values ±SD (n=4), ***p*<0.01, ****p*<0.001, *****p*<0.0001. **c.** Yield comparison of the target product **24a** in two timepoint (1 h and 4 h), which were obtained from A₁₂T₁₂C₁₃A*T₁₃ and its three mutants. The yield of **24a** generated from the A₁₂T₁₂C₁₃A*T₁₃ was quantified as a reference (100%), respectively. Data are presented as mean values ±SD (n=3).

10. Figure 4c legend, need to keep the residue numbers consistent with the full length protein.

Response: Corrected. We revised Figure 4 and its legend.

11. SI Figure 3, the phenylalanine ring was cut off in the structure of 9.

Response: We revised the structure of **9** and the structures of the other compounds.

12. In domain swapping experiments, were incomplete or shunt products formed? If so, at what levels? What inter domain-linkers were used? What were the isolated yields of the desired compounds in Figure 5e,f,g? Were the bioactivity tested on isolated compounds? Such information would be important to evaluate the efficiency of the NRPS engineering experiments.

Response: During the substitution domain process, we also detected some short intermediates (**28a**, **30a**, **30b**) that were hydrolyzed in advance from the assembly line in different recombinants, as shown in Supplementary Fig. 30. The substitution sites of *rzmA** and *holA*, along with their corresponding amino acid sequence and protein structure characteristics, have been summarized in the Supplementary Fig. 25. We isolated 3.6 mg mass of two derivatives **29** and **31** from 16 L and 12 L fermentation, respectively, which was described in method “Purification of glidonins and lipopeptide derivatives”. The derivative **32** was not isolated in this work due its low yield compared with **29** and **31**. As mentioned in line 358, the **29** and **31** derivatives showed moderate anti-inflammatory activities (LPS-induced NO production test). We also compared their antitumor activity with the group without Put moiety, and although they did not exhibit obvious antitumor activity ($IC_{50} > 40 \mu M$), both derivatives showed better overall bioactivity (anti-inflammatory and antitumor) than **29a** and **31a** without the Put moiety. (showed in Supplementary Table 12 and Fig. 29.). These results reveal the Put improved the bioactivity of lipopeptides as we suggested.

Supplementary Fig. 25. Swapping site of Put unit with *holA* and *rzmA** gene clusters. RzmA*-M1, HolA-M1, and HolA-M6 are swapped in site1, RzmA*-M2, HolA-M2, and HolA-M7 are swapped in site2, HolA-M3 and HolA-M9 are swapped in site1 and site3, HolA-M4 is swapped in site2 and site3, HolA-M8 is swapped in site1 and site4.

Supplementary Fig. 30. Yield comparison of the other intermediates from engineering *rzmA** and *holA*

BGCs between wild type and mutants. **a.** Yield comparison of **28** and the intermediate **28a** between wild type and two mutants (RzmA*-M1 and RzmA*-M2). The yield of **28** generated from the wild type strains was quantified as a reference (100%). **b, c.** Yield comparison of **30** and the intermediates **30a** and **30b** between wild type and eight mutants. HolA-M5 did not produce any products. The yield of **30** generated from the wild type strains was quantified as a reference (100%). **d.** Structures of C8-RzmA, Holrhizin A and their corresponding intermediates.

13. Figure 5h and 5i were barely discussed in the text with regard to the anti-inflammatory activities. The LPS-induced NO production was not explained at all.

Response: The engineering compounds **29** and **31** suppressed moderately the release of NO in LPS-induced RAW264.7 macrophages compared with **29a** and **31a** through the anti-inflammatory assays (Fig 5h and 5i). The related testing procedure was shown in method in line 772. The LPS-induced NO production was a method for testing the bioactivity of engineering compounds. According to the reviewer1's suggestion that this manuscript focuses on the biosynthesis and engineering, we moved the result of bioactivity of compounds to Supplementary Fig. 29.

Supplementary Fig. 29. The structures and anti-inflammatory bioactivity of compounds. **a.** Structures of lipopeptides **29a**, **29**, **31a**, and **31**. **b.** The effects of compounds on NO production in LPS-induced RAW264.7 cells between **29**, **31** and each Put-lacking derivatives **29a**, **31a**. NO production in the absence of LPS was used as negative control. LPS-stimulated NO production in the absence of added compounds was used as the positive control.

14. Figure 7c and d, what concentrations were the compounds?

Response: The concentrations of compounds were 0.1 $\mu\text{g}/\mu\text{L}$ and added it in method.

This paper has many spelling and grammatical mistakes. I've only listed a few examples:

1. Line 63-64, "amino acid moieties loaded in lipopeptides, which was proposed that this process might..." is awkward.

Response: We changed it to "The additional C-terminal moieties include the amino acid residues that might be mediated by the TE domain." We carefully changed some incorrect grammars, spellings and references, and the revised manuscript was then extensively edited by the Nature Research Editing Service, to improve the grammar, scientific accuracy, clarity and instructiveness of our manuscript.

2. Throughout the manuscript, lauricyl should be lauroyl, myristicyl should be myristoyl.

Response: We revised these mistakes in whole manuscript.

3. Line 114, define X.

Response: The "X" indicates unknown amino acid, we added it into the manuscript.

4. Line 119, should be "ten conserved active site residues"

Response: Thank you for your advice. We changed this sentence to "The absence of the N-terminal subdomain (A_{core}) and the Stachelhaus code in the A^* domain leads to the A^* domain retaining only a C-terminal subdomain (A_{sub}), which means it fails to function in amino acid activation."

5. Line 129, "revealed that 1 was confirmed to be" is awkward.

Response: Corrected. We changed it to "revealed that **1** was a dodecapeptide containing the following amino acid moieties."

6. Line 205, "opened" should be "open"

Response: Corrected. We moved this part to the supplementary part according to the review¹ suggestion, which was “Supplementary Note²: The substrate promiscuity of acylase GdnD”. The sentence was changed to “At first, we predicted the structure of GdnD using AlphaFold²¹, which is “V”-shaped structure, and its substrate pocket is open that can accommodate broad substrates”

7. Line 274, “to furtherly clarify”

Response: We changed it to “To gain more insight into”

8. Line 307-309, confusing sentence.

Response: We revised this section clearly, and the corresponding revisions were marked in blue color. This sentence was changed to “Based on the aforementioned results, we can infer that the C₁₃ domain is directly involved in the intermolecular condensation of T-linked Ala and free Put. Moreover, the A* and T₁₃ domains are essential for protein folding and stability *in vitro*.”

REVIEWER COMMENTS

Reviewer #1 (Remarks to the Author):

The authors have addressed all issues that I raised thoroughly. I thank them for going the extra mile and doing the suggested protein-MS work on the A1T1 didomain to show amino acid loading. Edits to the figures are exactly as requested – I hope the authors agree these are now greatly improved.

I congratulate them on an excellent piece of work!

Reviewer #2 (Remarks to the Author):

The paper by Chen et al has been revised quite a bit and I find it much better now. I am also fine with most of the responses to my and all other reviewer's responses.

However, I still have a few points which must be addressed:

- regarding the A* domain: What I was asking for is, to what A domain the A* domain is most similar to. Can this be determined using a BLAST analysis of the A* domain sequence against the producer or other bacteria in general (and then what is the specificity of the identified A domain then)?
- Fig. 1.: Where is GdnD located relatively to the Gdn BGC? Please add this information to Fig. 1a
- Fig. 5/S30/S32: In my opinion the quantification in Fig 5c, 5f and 5g and Suppl Fig 30b/c based on just one compound with all derivatives relative to the one is problematic. The differences are in the attachment of putrescine resulting in much better ionization and even comparing cyclic with linear plus putrescin; linear always ionizes better than cyclic even without putrescin already. Since in Supp Fig 33 absolute amounts are shown (ng), why was something like this not used in the other figures? At least a linear with/without putrescin and cyclic derivative must be used as quantification standard.
- structures: I appreciate the much better drawing of all structures but especially in the MS figures, several structures are still not linear and more difficult to read (S9-S14). For a good example see S15.
- Supp Fig 28 still needs MS annotations in structures or spectra
- Supp Fig 30: I would suggest to add the yield to 30a in Fig. S30c or more the broken axis more between 50-100%
- NMR spectra: add field strength to ALL NMR spectra as well.
- NMR tables: Supp Table 6: MHz not M

Reviewer #3 (Remarks to the Author):

The revised manuscript addressed most of my comments. A few remaining concerns:

1. Figure 1, the arrow left to putrescine should be a straight reaction arrow.
2. Cd is not the right nomenclature for dual C. It should be C_{dual} (dual subscript), or dual C/E.
3. Considering that TE1 and TE2 are two distinct full TE domains, I recommend illustrating TE1 and TE2 as two domains instead of one in the figures throughout for clarity.
4. Line 437-438: mixed up sentence, "thus, acylated glidonins could improve biofilm formation by inhibiting the swarming motility of DSM 7029 led to alter to colony morphology of DSM 7029"
5. Fig 4: the H9721A mutant of C13 did not abolish production of 24a, it reduces it to ~20%. The description in line 225 is inaccurate.

REVIEWER COMMENTS

Reviewer #1 (Remarks to the Author):

The authors have addressed all issues that I raised thoroughly. I thank them for going the extra mile and doing the suggested protein-MS work on the A1T1 didomain to show amino acid loading. Edits to the figures are exactly as requested – I hope the authors agree these are now greatly improved.

I congratulate them on an excellent piece of work!

Response: We sincerely appreciate the reviewer's positive comments and recognition of our work.

Reviewer #2 (Remarks to the Author):

The paper by Chen et al has been revised quite a bit and I find it much better now. I am also fine with most of the responses to my and all other reviewer's responses.

However, I still have a few points which must be addressed:

- regarding the A* domain: What I was asking for is, to what A domain the A* domain is most similar to. Can this be determined using a BLAST analysis of the A* domain sequence against the producer or other bacteria in general (and then what is the specificity of the identified A domain then)?

Response: The partial A* domain includes only 190 amino acid residues, which the intact A domain usually contains ~ 400 AAs. The A* domain shows low coverage (190/409) and moderate identities (41%-45%) to the A domains in the non-ribosomal peptide synthetases in the producer DSM 7029 strain by BlastP analysis (Supplementary Table 15, Supplementary Fig. 19d).

Based on the result of blastP analysis, we still could not ensure the substrate specificity of the A* domain because the A* was similar to various A domains in the producer. The ten Stachelhaus codes in the amino acid binding pocket for determining the specificity of A domain are absent in the A* domain. Thus, the specificity of the retained fragment A* domain was unknown based on the BlastP analysis and prediction. The description “The A* domain shows similarity (Coverage: 190/409; Identities: 41%-45%) to other A domains of GdnB, but it is not possible to predict the specificity of the A* domain due to the absence of the Stachelhaus codes (Supplementary Table 15, Supplementary Fig. 19d).” was added in the manuscript.

Supplementary Table 15. The identity between GdnB-A* and other A domains of NRPSs in *Schlegelella brevitalea* DSM 7029 by the BlastP analysis

Domain	Coverage (aa)	Identities (aa)	Substrate
GdnB-A*/GdnB-A6	190/409	50/111 (45%)	Pro
GdnB-A*/GdnB-A8	190/398	48/106 (45%)	Val
GdnB-A*/GdnB-A7	190/404	49/111 (44%)	Trp

GdnB-A*/GdnB-A12	190/394	47/106 (44%)	Ala
GdnB-A*/GdnB-A4	190/397	47/106 (44%)	Val
GdnB-A*/GdnB-A9	190/394	46/106 (43%)	Ala
GdnB-A*/GdnB-A10	190/394	48/106 (45%)	Ala
GdnB-A*/GdnB-A11	190/413	47/111(42%)	Ser
GdnB-A*/GdnB-A5	190/400	46/111(41%)	Phe

Supplementary Fig. 19. Bioinformatic analysis of Gdn NRPS gene cluster. **a.** AntiSMASH prediction of Gdn NRPS gene cluster. **b.** BlastP analysis of the C₁₃A*T₁₃ domains of the GdnB. **c.** Sequence alignment of GdnB-A* and GrsA(1AMU_A). The red frame indicates Stachelhaus code. **d.** Sequence alignment of GdnB-A* and A domains of GdnB (WP_053013656.1) of the producer *Caldimonas brevitalea* (previously identified as *Schlegella brevitalea* DSM 7029) based on blastP analysis. Alpha helix in magenta and β -strands in yellow.

- Fig. 1.: Where is GdnD located relatively to the Gdn BGC? Please add this information to Fig. 1a

Response: The distance between *gdnD* and *gdn* BGC was 1,002,361bp, which was shown in the Fig. 1a, and the description “The deletion of AKJ30942.1 (named GdnD), which showed 68% similarity to PvdQ and was far away from the *gdn* BGC (1,002,342 bp), abolished the production of **1-5**, as shown by HPLC–MS analysis” was added in the manuscript.

Fig 1. The biosynthetic pathway of glidonins. (a) Genetic organization of the *gdn* gene cluster. (b) Organization of modular enzymes involved in the biosynthesis of glidonin. (c) Structures and formation of glidonins A-L (1-12). FMT: methionyl-tRNA formyltransferase, Cs: starter condensation, C_{dual}: dual C/E, C: condensation, A: adenylation, T: thiolation, TE: thioesterase, A*: partial adenylation.

- Fig. 5/S30/S32: In my opinion the quantification in Fig 5c, 5f and 5g and Suppl Fig 30b/c based on just one compound with all derivatives relative to the one is problematic. The differences are in the attachment of putrescine resulting in much better ionization and even comparing cyclic with linear plus putrescine; linear always ionizes better than cyclic even without putrescine already.

Since in Suppl Fig 33 absolute amounts are shown (ng), why was something like this not used in the other figures? At least a linear with/without putrescine and cyclic derivative must be used as quantification standard.

Response: We revised Fig. 5 and Fig. S30 using the absolute yield of the related compounds determined by the standard curve of **28**, **29a**, **29**, **30a** and **31** according to your suggestion, respectively. The detailed description “In addition, the derivatives and intermediates (or original products) produced by the engineering *rzmA** and *holA* BGCs

were quantified using the standard curve of compounds **28** (for quantification of compound **28**), **29a** (for quantification of compound **28a**), **29** (for quantification of compound **29**), **30a** (for quantification of compounds **30**, **30a**, **30b**) and **31** (for quantification of compound **31**) using HPLC-MS measurement as described above (5%-95% ACN in 15 min). Triplicates of all experiments were measured.” was added in method.

Fig 5. Introduction of putrescine moiety on the two NRPs by multi-domain swapping. (a) The flow chart of engineering *rzmA** BGC and corresponding products. (b) A diagram of the different multi-domain substitutions of the C-terminus of RzmA-R148A (RzmA*). (c) Yield comparison of C8-Rzm A (**28**) and its derivative **29** between wild type (WT) and mutants. The absolute yield of related compounds was determined using the standard curve of **28** and **29**, respectively. (d) The flow chart of engineering *holA* BGC and corresponding products. (e) A diagram of the different multi-domain substitutions of the C-terminus of holrhizin A. (f)-(g) Yield comparison of holrhizin A (**30**) as well as their derivatives **31** and **32** between WT and mutants. The absolute yield of related compounds was determined using the standard curve of **30a** and **31**, respectively. N.D. indicates no products. Data are presented as mean values \pm SD (n=3). *p* values were determined by two-tailed unpaired *t* test. *****p*<0.0001. Source data and the exact *p* values are provided as a Source Data file.

Supplementary Fig. 30. Yield comparison of the other intermediates from engineering *rzmA** and *hola* BGCs between wild type and mutants. **a.** Yield comparison of **28** and the intermediate **28a** between wild type and two mutants (RzmA*-M1 and RzmA*-M2). The absolute yield of related compounds was determined using the standard curve of **28** and **29a**, respectively. **b, c.** Yield comparison of **30** and the intermediates **30a** and **30b** between wild type and eight mutants. HolA-M5 did not produce any products. The absolute yield of related compounds was determined using the standard curve of **30a**. **d.** Structures of C8-Rzm A, holrhizin A and their corresponding intermediates. N.D. indicates no products.

- structures: I appreciate the much better drawing of all structures but especially in the MS figures, several structures are still not linear and more difficult to read (S9-S14). For a good example see S15.

Response: We revised all structures of the figures in the whole manuscript.

- Supp Fig 28 still needs MS annotations in structures or spectra

Response: Revised as suggested.

Supplementary Fig. 28. HR-ESI-MS spectra and MS/MS fragmentation of **13**, **14** and **15**.

- Supp Fig 30: I would suggest to add the yield to 30a in Fig. S30c or more the broken axis more between 50-100%.

Response: We revised the Fig. S30c using the absolute yield of related compounds **30**, **30a** and **30b** based on the standard curve of **30a** according to your suggestion.

- NMR spectra: add field strength to ALL NMR spectra as well.

Response: Revised as suggested.

- NMR tables: Supp Table 6: MHz not M

Response: Revised as suggested.

Reviewer #3 (Remarks to the Author):

The revised manuscript addressed most of my comments. A few remaining concerns:

1. Figure 1, the arrow left to putrescine should be a straight reaction arrow.

Response: Revised as suggested.

Fig 1. The biosynthetic pathway of glidonins. (a) Genetic organization of the *gdn* gene cluster. **(b)** Organization of modular enzymes involved in the biosynthesis of glidonin. **(c)** Structures and formation of glidonins A-L (1-12). FMT: methionyl-tRNA formyltransferase, Cs: starter condensation, C_{dual}: dual condensation (C/E), C: condensation, A: adenylation, T: thiolation, TE: thioesterase, A*: partial adenylation.

2. Cd is not the right nomenclature for dual C. It should be Cdual (dual subscript), or dual C/E.

Response: We revised Fig. 1 and Fig. 5 as your suggestion.

3. Considering that TE1 and TE2 are two distinct full TE domains, I recommend illustrating TE1 and TE2 as two domains instead of one in the figures throughout for clarity.

Response: Revised as suggested.

4. Line 437-438: mixed up sentence, “thus, acylated glidonins could improve biofilm formation by inhibiting the swarming motility of DSM 7029 led to alter to colony morphology of DSM 7029”

Response: We have corrected the mixed-up sentence as follows: "Thus, acylated glidonins could improve biofilm formation by inhibiting the swarming motility of DSM 7029, leading to alterations in the colony morphology of DSM 7029."

5. Fig 4: the H9721A mutant of C₁₃ did not abolish production of 24a, it reduces it to ~20%. The description in line 225 is inaccurate.

Response: At line 225 in the previous manuscript, the H9271A mutant of C₁₃ indeed abolished production of glidonins *in vivo* assay that was shown in Fig. 3d. But, by *in vitro* chemical assay, the H9271A mutant of C₁₃ reduced production of **24a** (~20%) that was shown in Fig. 4d. Hence, mutating the C₁₃ domain caused different effect to production of targeted product (glidonins or **24a**) through the *in vivo* and *in vitro* assays. The *in vitro* result suggested that the free amino of Put could still attack the carbonyl of T₁₂-linked Ala to form Ala-Put (**24**) in the group of mutant H9271A. While, the H9271A mutant of C₁₃ domain might greatly affect the NRPS GdnB leading to abolish the production of glidonins.

REVIEWERS' COMMENTS

Reviewer #2 (Remarks to the Author):

All my points have now been addressed by the authors. Well done and a nice piece of work.